# BIOCLIP 2: Emergent Properties from Scaling Hierarchical Contrastive Learning

**Jianyang Gu**[1†], **Samuel Stevens**[1], **Elizabeth G Campolongo**[1], **Matthew J Thompson**[1],
**Net Zhang**[1], **Jiaman Wu**[1], **Andrei Kopanev**[1], **Zheda Mai**[1], **Alexander E. White**[2],
**James Balhoff**[3], **Wasila Dahdul**[4], **Daniel Rubenstein**[5], **Hilmar Lapp**[6],
**Tanya Berger-Wolf**[1], **Wei-Lun Chao**[1], **Yu Su**[1†]

[1]The Ohio State University, [2]Smithsonian Institution, [3]UNC Chapel Hill,
[4]University of California, Irvine, [5]Princeton University, [6]Duke University

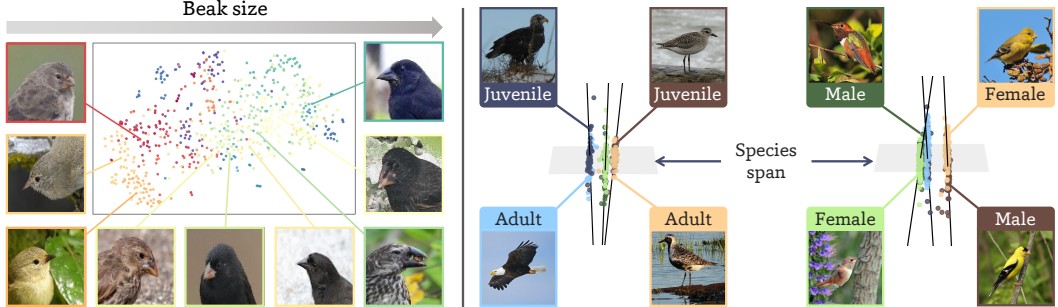

Figure 1: While BIOCLIP 2 is trained to distinguish species, it demonstrates emergent properties beyond the initial training objective. **Left**: At the *inter-species* level, the embedding distribution of different species aligns with ecological relationships; the embeddings of Darwin's finches arrange themselves by beak size from left to right. **Right**: Instead of collapsing, the *intra-species* variations are preserved in subspaces orthogonal to the inter-species variation (the black lines point from the mean embedding of one variant to that of the other variant). Orthogonality increases with scale (see Figure 3c).

## Abstract

Foundation models trained at scale exhibit remarkable emergent behaviors, learning new capabilities beyond their initial training objectives. We find such emergent behaviors in biological vision models via large-scale contrastive vision-language training. To achieve this, we first curate TREEOFLIFE-200M, comprising 214 million images of living organisms, the largest and most diverse biological organism image dataset to date. We then train BIOCLIP 2 on TREEOFLIFE-200M to distinguish different species. Despite the narrow training objective, BIOCLIP 2 yields extraordinary accuracy when applied to various biological visual tasks such as habitat classification and trait prediction. We identify emergent properties in the learned embedding space of BIOCLIP 2. At the inter-species level, the embedding distribution of different species aligns closely with functional and ecological meanings (*e.g.*, beak sizes and habitats). At the intra-species level, instead of being diminished, the intra-species variations (*e.g.*, life stages and sexes) are preserved and better separated in subspaces orthogonal to inter-species distinctions. We provide formal proof and analyses to explain why hierarchical supervision and contrastive objectives encourage these emergent properties. Crucially, our results reveal that these properties become increasingly significant with larger-scale training data, leading to a biologically meaningful embedding space.

Models, data, and code available at imageomics.github.io/bioclip-2. [†]{gu.1220, su.809}@osu.edu

39th Conference on Neural Information Processing Systems (NeurIPS 2025).

# 1 Introduction

Recent advances in artificial intelligence (AI) are transforming core scientific workflows to become more efficient and automated [1, 2]. Tasks that once demanded overwhelming time and labor, like inferring atomic protein folds, designing functional materials, or producing global weather forecasts, can now be efficiently accomplished by large-scale predictive models [3, 4, 5, 6, 7, 8]. Particularly, a growing class of *domain-specific foundation models* demonstrate capabilities that arise without explicit definition during training [9, 10]. For example, language models trained purely on large-scale amino-acid strings unexpectedly develop an understanding of 3D chemistry to predict atomic folds with near-experimental accuracy [4, 11]. These scale-driven emergent abilities are reshaping scientific inference and opening new avenues for data-centric discovery.

In ecology and evolutionary biology, previous efforts leveraged hierarchical taxonomic labels and CLIP-style contrastive training [12] to achieve pronounced species classification accuracy across the tree of life [13, 14, 15]. This work asks a simple but intriguing question: *what properties emerge if we scale up hierarchical contrastive training?* To answer this question, we curate TREEOFLIFE-200M, comprising 214M organism images spanning 952K taxonomic classes, making it the largest and most diverse visual catalog of life to date. Through training at scale, our model BIOCLIP 2 improves species classification accuracy by 18.0% over BIOCLIP [13]. More importantly, we explore whether representations learned solely through species-level supervision can generalize to diverse biological questions **beyond species classification**.

To probe these capabilities, we evaluate BIOCLIP 2 on a variety of existing biological visual tasks, including habitat classification [16], trait prediction [17, 18], new-species identification [19], and agricultural disease detection [20]. These applications push beyond simple species recognition and apply directly to biodiversity conservation, trait organization, and agricultural health. Despite being trained primarily with species-level supervision, BIOCLIP 2 outperforms both vision-language (*e.g.*, SigLIP [21]) and vision-only baselines (*e.g.*, DINOv3 [22]) by an average margin of 10.3% on these tasks. We then look deeper into the embedding space of BIOCLIP 2 and identify two *emergent properties* as the training scales up.

At the **inter-species** level, the embedding distribution of different species aligns with their ecological relationships. As shown on the left side of Figure 1, BIOCLIP 2 embeddings of Darwin's finches demonstrate an increasing beak size from left to right, which is not observed in the original CLIP embedding space. We attribute the property to the adopted hierarchical taxonomic labels, which inherently encode functional and ecological information [23]. The hierarchical supervision at scale pushes related species to co-locate in functionally coherent "macro-clusters." In such a way, BIOCLIP 2 acquires functional trait knowledge without using explicitly labeled traits.

At the **intra-species** level, contrary to the intuition that fine-grained differences collapse after extensive training [24, 25], BIOCLIP 2 keeps the intra-species variations (*e.g.*, life stages and sexes) distinct. On the right side of Figure 1, three species form tight clusters when projected onto the "species plane," while their intra-species variations fan out along axes orthogonal to the plane. Such variation cues are not encoded in taxonomic labels. We theoretically prove that when species prototypes are nearly orthogonal (*i.e.*, species are well separated), the contrastive objective prioritizes orthogonality between intra-species variations and inter-species differences over raw magnitude (Theorem 5.1). Furthermore, these variations are observed to be increasingly separable as training data scales up. This microstructure preserves the intra-species representational diversity without interfering with inter-species distinctions, enabling various attribute recognition applications (§5.1).

We show in §5.2 through quantitative and qualitative analyses that larger-scale training improves both inter-species ecological alignment and separation of intra-species variants. These scale-amplified patterns make the embedding space more interpretable and biologically meaningful. BIOCLIP 2 evidences that combining domain-specific scaling with structured supervision can unlock qualitatively new emergent behaviors in scientific vision models.

# 2 Related Work

**Emergent properties in foundation models.** Emergent properties refer to the capabilities implicitly acquired from the training process and generalized beyond the initial training objective. Large language models (LLMs) illustrate a variety of in-context learning skills after next-token-prediction

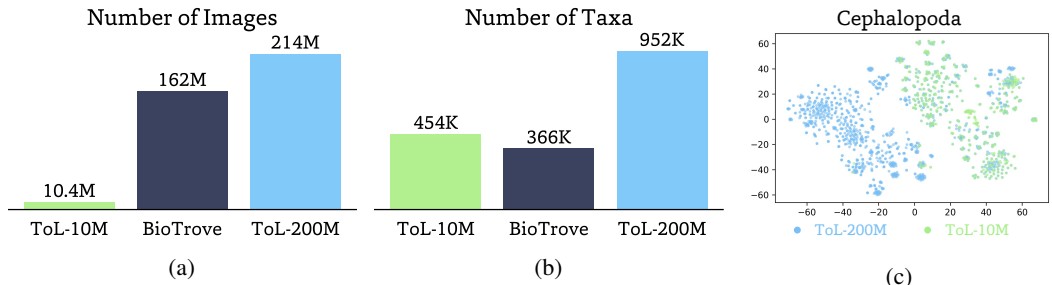

Figure 2: **(a)** Number of images across organismal biology datasets. **(b)** Biodiversity comparison across datasets (measured unique 7-tuples for TREEOFLIFE (ToL) datasets, species count provided by BioTrove). **(c)** The taxa distributional difference in the *Cephalopoda* class (octopuses, squids, *etc*.) between ToL-200M and ToL-10M.

pre-training [9, 26]. Such emergence also arises in computer vision [27]. DINO learns semantic segmentation through purely visual self-supervision [28], whereas GroupViT learns semantic segmentation solely through text supervision [29]. The studies closest to this work are [30, 31]. Alper and Averbuch-Elor explore the visual-semantic hierarchies in CLIP models [30]. Although hierarchical semantic structures are never explicitly presented as the training supervision, CLIP models acquire the capability of matching images with varying levels of descriptions. Abbasi *et al*. discover that CLIP models possess disentangled representational sub-spaces for different factors of variations [31]. It allows CLIP models to generalize across compositional out-of-distribution concepts. This work leverages CLIP to distinguish different species with hierarchical labels, which is a different scenario from the above work. We investigate the emergent properties under this setting.

**Computer vision for ecology & evolutionary biology.** Ecology and evolutionary biology are naturally challenging for computer vision systems due to long-tail distributions, extremely fine-grained classifications, and a wide variety of image distributions. Existing work formalizes these challenges into specific visual tasks such as attribute prediction [NeWT, 18], trait prediction [FishNet, 16], and plant disease detection [PlantDoc, 20].

Recent advancements in computer vision have led to the development of foundation models for biological applications. BIOCLIP incorporates taxonomic labels in the vision-language contrastive training, yielding promising species classification accuracy [13]. Follow-up work scaled data to 162M images [BioTrove, 14], specialized the data to camera traps [CATALOG and WildCLIP, 32, 33], and added additional model modalities [TaxaBind, 34]. We investigate both data and model scaling, with a focus on both broad biological applications and any emergent properties after extensive training.

## 3 TREEOFLIFE-200M

Large-scale, clean, diverse data drives progress in machine learning. There have been efforts such as TREEOFLIFE-10M [35] and BioTrove [14] to create large-scale biological organismal datasets for machine learning (ML). As shown in Figure 2, TREEOFLIFE-10M [35] improves on prior work such as iNat21 [36] and BIOSCAN-1M [37] by increasing taxa diversity by a factor of 45. BioTrove [14] increases data scale to 162M but fails to match the biodiversity of TREEOFLIFE-10M. In this work, we combine the vast breadth of Global Biodiversity Information Facility (GBIF) [38] images with those of the Encyclopedia of Life project (EOL) [39], BIOSCAN-5M [40], and FathomNet Database [41]. With nearly 214 million images representing 952,257 taxa, TREEOFLIFE-200M is the largest *and* most diverse public ML-ready dataset for computer vision models in biology.

Unlike BioTrove [14], which relies solely on iNaturalist and contains 162M images but only 366K unique species, our use of museum, camera-trap, and citizen-science contributions expands the taxonomic coverage to $2.6\times$ more taxa. Our curation efforts ensure this breadth does not come at the cost of data quality. We quantify image type diversity from GBIF: 51.8M museum specimen, 617.8K camera trap, and 151M citizen science images. Beyond the taxa-wise diversity, these images also provide more observing perspectives for the focal species. In such a way, the robustness of models trained on TREEOFLIFE-200M is significantly enhanced against a variety of use cases. Specifically, we demonstrate that BIOCLIP 2 yields a $22.8\%$ performance gap compared with BIOCLIP on camera trap images (See Table 1). We provide detailed statistics of the image number and taxa diversity from each data provider in Figure 11.

## 3.1 Images

TREEOFLIFE-200M consists of images curated from four core data providers: GBIF, EOL, BIOSCAN-5M, and FathomNet. GBIF and EOL aggregate biological data from various sources, such as iNaturalist [42], the Smithsonian Institution [43], and Flickr [44]. BIOSCAN-5M and FathomNet are curated collections of expert-annotated images designed to improve cataloging and identification of species from highly diverse and under-represented branches of the tree of life. BIOSCAN-5M is part of the ongoing DNA Barcoding project to improve insect identification (one of the most diverse classes, *Insecta*). FathomNet focuses on a habitat rather than a clade: all animals that live in the ocean. Together, these comprise TREEOFLIFE-200M.

## 3.2 Data Curation & Filtering

We retrieve data from data providers using `distributed-downloader` [45]. The initial retrieval gives us 222,065,140 images and 1,359,405 unique taxonomic hierarchies. We then cleaned the data, focusing on (1) aligning taxonomic labels, (2) image quality, and (3) eliminating duplication and data leakage. We summarize these efforts here, with more details provided in §I.

**Taxonomic alignment.** Curating large biological image collections from distributed data providers requires the alignment of noisy and inconsistent taxonomic labels both between and within providers. We develop a taxonomic alignment package `TaxonoPy` in consultation with taxonomists that resolves entries to both a seven-rank Linnaean hierarchy and a common name. `TaxonoPy` queries GNVerifier [46] against the GBIF Backbone, Catalogue of Life, and OpenTree hierarchies (in that order). From the original 1.36M taxa, `TaxonoPy` filters 407K taxonomic hierarchies (such as synonyms, provisional names, etc.) and yields 952K unique taxa.

**Image-quality screens.** Digital archives contain herbarium labels, empty camera-trap frames, and occasional people. None add biological signal, and faces raise privacy concerns, so we drop them to keep learning focused on organisms via three neural-network-based filters: (i) **Museum non-organism removal.** A pre-trained CLIP-L/14 [12] is used for a nearest-centroid classifier spanning 25 fine-grained subtypes (10 collection areas, each split into categories such as specimen, fossil, drawer-label, etc.). The classifier is fit to 8.5K manually-curated examples and predicts a subtype for all museum images; we drop all non-organismal images. (ii) **Camera-trap trimming.** We apply a pre-trained camera-trap model MegaDetector [47, 48] to filter for frames with visible animals. (iii) **Face removal.** We apply a pre-trained face-detection model MTCNN [49] to discard images containing human faces. We release the code used in processing the images in `TreeOfLife-toolbox`.

**Duplicate and leakage control.** Exact duplicates in the training set are removed with MD5 hashes. GBIF includes images from iNaturalist, a popular source for computer vision ecology benchmarks. To prevent train-test leakage and inflated downstream scores, we compute both MD5 and perceptual PDQ [50] hashes for every test image and purge any near or exact duplicates from training.

## 3.3 Taxa Coverage

The International Union for Conservation of Nature (IUCN) estimates 2.14M species have been described [51]. Following curation, there are nearly 868K unique taxa labeled to the level of species in TREEOFLIFE-200M.[1] Based on the most recent IUCN Red List assessment [52], TREEOFLIFE-200M demonstrates a particularly strong representation of threatened species, with 77.1% coverage (36,370 species). This coverage establishes that the approach to integrating diverse data sources used in TREEOFLIFE-200M is a valuable resource for conservation research, providing representation for a substantial majority of species prioritized for global conservation action.

Notably, TREEOFLIFE-200M adds diverse clades that are extremely under-represented in prior work like TREEOFLIFE-10M and BioTrove. Figure 2c compares the distribution of taxonomic names in the class of *Cephalopoda* between TREEOFLIFE-10M and TREEOFLIFE-200M. While they share overlaps, there are clades almost completely absent in TREEOFLIFE-10M. These under-represented clades receive a substantial influx of samples in TREEOFLIFE-200M (1,102 new taxa). Another

---

[1]Not all images contain full 7-rank Linnaean taxa; for instance, 93% of BIOSCAN-5M images are not labeled to the species level. Thus, the unique taxa with non-null species is a more appropriate comparison.

Table 1: Zero-, one-, and five-shot species classification top-1 accuracy across 10 tasks for different models. **Bold** and underlined entries indicate the **best** and second best accuracies, respectively. BIOCLIP 2 outperforms both strong general- (CLIP, SigLIP, DINOv3) and domain-specific- (BIOCLIP, BioTrove-CLIP) baselines. "Camera Trap" is mean performance across 5 camera-trap datasets; Appendix F.4 contains more details.

| | Animals | | | | | Plants & Fungi | | | | | |
| Model | NABirds | Plankton | Insects | Insects 2 | Camera Trap | PlantNet | Fungi | PlantVillage | Med. Leaf | Rare Species | Mean |
|---|---|---|---|---|---|---|---|---|---|---|---|
| Random Guessing | 0.2 | 1.2 | 1.0 | 1.0 | 3.5 | 4.0 | 4.0 | 2.6 | 4.0 | 0.3 | 2.2 |
| *Zero-Shot Classification* | | | | | | | | | | | |
| CLIP (ViT-L/14) | 66.5 | 1.3 | 9.0 | 11.7 | 29.5 | 61.7 | 7.6 | 6.5 | 25.6 | 35.2 | 25.5 |
| SigLIP | 61.7 | 2.4 | 27.3 | 20.7 | 34.1 | 81.8 | 36.9 | **28.5** | 54.5 | 47.6 | 39.6 |
| BioTrove-CLIP | 39.4 | 1.0 | 20.5 | 15.7 | 11.0 | 64.4 | 38.2 | 15.7 | 31.6 | 24.6 | 26.2 |
| BIOCLIP | 58.8 | **6.1** | 34.9 | 20.5 | 31.8 | 88.2 | 40.9 | 19.0 | 38.5 | 37.1 | 37.6 |
| BIOCLIP 2 | **74.9** | 3.9 | 55.3 | 27.7 | 53.8 | **96.8** | **83.8** | 25.1 | **57.8** | **76.8** | **55.6** |
| *One-Shot Classification* | | | | | | | | | | | |
| CLIP (ViT-L/14) | 42.7$_{\pm0.8}$ | 28.9$_{\pm0.6}$ | 29.0$_{\pm0.4}$ | 17.0$_{\pm0.8}$ | 36.0$_{\pm2.8}$ | 58.7$_{\pm2.8}$ | 20.7$_{\pm2.2}$ | 56.7$_{\pm2.1}$ | 74.4$_{\pm1.9}$ | 34.3$_{\pm0.8}$ | 39.8 |
| SigLIP | 39.9$_{\pm0.9}$ | 28.4$_{\pm0.5}$ | 32.3$_{\pm0.6}$ | 20.6$_{\pm1.3}$ | 37.8$_{\pm2.6}$ | 66.3$_{\pm5.3}$ | 28.7$_{\pm0.9}$ | 64.1$_{\pm3.0}$ | 81.7$_{\pm2.3}$ | 38.8$_{\pm0.7}$ | 43.9 |
| Supervised-IN21K | 43.8$_{\pm0.8}$ | 23.5$_{\pm1.2}$ | 15.2$_{\pm1.1}$ | 18.2$_{\pm1.5}$ | 30.6$_{\pm2.7}$ | 63.8$_{\pm4.4}$ | 26.4$_{\pm1.4}$ | 52.8$_{\pm3.5}$ | 75.2$_{\pm4.4}$ | 31.6$_{\pm0.7}$ | 38.1 |
| DINOv3 | 48.3$_{\pm1.1}$ | **36.5**$_{\pm0.8}$ | 8.8$_{\pm0.7}$ | 18.8$_{\pm1.6}$ | 43.0$_{\pm2.7}$ | 66.8$_{\pm4.7}$ | 27.5$_{\pm1.8}$ | 64.8$_{\pm1.2}$ | **92.1**$_{\pm2.2}$ | 41.5$_{\pm0.2}$ | 44.8 |
| BioTrove-CLIP | 61.9$_{\pm0.6}$ | 26.4$_{\pm0.5}$ | 57.1$_{\pm1.4}$ | 20.9$_{\pm0.7}$ | 31.2$_{\pm2.3}$ | 69.7$_{\pm3.4}$ | 47.3$_{\pm2.1}$ | 55.8$_{\pm3.4}$ | 83.5$_{\pm1.1}$ | 34.9$_{\pm0.4}$ | 48.9 |
| BIOCLIP | 57.4$_{\pm1.2}$ | 29.7$_{\pm1.1}$ | 57.1$_{\pm1.0}$ | 20.4$_{\pm0.9}$ | 35.0$_{\pm2.8}$ | 67.7$_{\pm3.9}$ | 44.6$_{\pm2.0}$ | 59.5$_{\pm2.5}$ | 83.7$_{\pm1.8}$ | 44.9$_{\pm0.7}$ | 50.0 |
| BIOCLIP 2 | **82.4**$_{\pm1.1}$ | 32.0$_{\pm0.4}$ | **74.6**$_{\pm0.4}$ | **28.4**$_{\pm0.7}$ | **48.1**$_{\pm2.2}$ | **85.8**$_{\pm4.5}$ | **70.3**$_{\pm2.6}$ | **67.6**$_{\pm1.1}$ | 92.0$_{\pm1.9}$ | **59.5**$_{\pm0.9}$ | **64.1** |
| *Five-Shot Classification* | | | | | | | | | | | |
| CLIP (ViT-L/14) | 68.2$_{\pm0.3}$ | 48.2$_{\pm1.5}$ | 50.6$_{\pm0.7}$ | 30.1$_{\pm0.7}$ | 53.9$_{\pm2.2}$ | 75.9$_{\pm1.2}$ | 31.4$_{\pm2.5}$ | 78.3$_{\pm1.4}$ | 92.6$_{\pm0.7}$ | 53.3$_{\pm0.4}$ | 58.2 |
| SigLIP | 64.2$_{\pm0.3}$ | 47.4$_{\pm1.1}$ | 54.9$_{\pm0.7}$ | 35.2$_{\pm0.5}$ | 56.9$_{\pm2.0}$ | 81.6$_{\pm1.4}$ | 45.5$_{\pm1.6}$ | 81.1$_{\pm0.7}$ | 94.1$_{\pm0.7}$ | 57.8$_{\pm0.6}$ | 61.9 |
| Supervised-IN21K | 57.5$_{\pm0.4}$ | 40.1$_{\pm0.6}$ | 30.1$_{\pm0.7}$ | 30.3$_{\pm0.2}$ | 48.3$_{\pm2.6}$ | 77.2$_{\pm1.4}$ | 39.6$_{\pm1.9}$ | 78.0$_{\pm1.1}$ | 92.8$_{\pm0.9}$ | 48.8$_{\pm0.4}$ | 54.3 |
| DINOv3 | 72.3$_{\pm0.4}$ | **57.8**$_{\pm1.6}$ | 19.7$_{\pm0.7}$ | 34.2$_{\pm0.6}$ | 63.6$_{\pm2.8}$ | 83.3$_{\pm1.4}$ | 44.1$_{\pm1.6}$ | 82.4$_{\pm1.1}$ | **98.8**$_{\pm0.5}$ | 62.6$_{\pm0.5}$ | 61.9 |
| BioTrove-CLIP | 78.5$_{\pm0.2}$ | 44.6$_{\pm0.6}$ | 77.0$_{\pm0.8}$ | 34.2$_{\pm0.6}$ | 47.9$_{\pm2.0}$ | 86.0$_{\pm1.0}$ | 65.2$_{\pm0.8}$ | 75.1$_{\pm0.8}$ | 96.2$_{\pm0.7}$ | 51.3$_{\pm0.2}$ | 65.6 |
| BIOCLIP | 78.2$_{\pm0.3}$ | 49.2$_{\pm1.1}$ | 78.0$_{\pm0.6}$ | 33.9$_{\pm0.6}$ | 54.3$_{\pm2.2}$ | 85.7$_{\pm1.7}$ | 61.6$_{\pm1.9}$ | 81.7$_{\pm1.1}$ | 96.7$_{\pm0.6}$ | 65.7$_{\pm0.4}$ | 68.5 |
| BIOCLIP 2 | **92.4**$_{\pm0.2}$ | 50.5$_{\pm1.1}$ | **89.3**$_{\pm0.4}$ | **44.3**$_{\pm1.1}$ | **67.7**$_{\pm1.9}$ | **94.4**$_{\pm0.8}$ | **85.0**$_{\pm1.1}$ | **83.9**$_{\pm0.9}$ | 98.4$_{\pm0.4}$ | **77.2**$_{\pm0.4}$ | **78.3** |

example is 55,085 taxa of *Fungi* in TREEOFLIFE-200M, close to $4\times$ of that in TREEOFLIFE-10M (14,793). The improved diversity facilitates accurate species classification of these clades, as evidenced in Table 1 (42.9% absolute improvement over BIOCLIP on zero-shot Fungi benchmark).

## 4 BIOCLIP 2 and Species Classification

BIOCLIP adopts a hierarchical multi-modal contrastive training framework, where images are associated with their corresponding hierarchical labels including taxonomic labels, scientific names, and common names [13]. Different from one-hot labels, taxonomic labels inherently encode hierarchical biological information from different levels [15]. In combination with an auto-regressive text encoder, BIOCLIP yielded superior species classification performance on both zero- and few-shot settings. In this work, we stick with the hierarchical contrastive training recipe and focus on the impact of scale.

**Modifications.** In addition to the significantly larger and more diverse dataset, we also scale model capacity by adopting a larger vision transformer (ViT-L/14 pre-trained on LAION-2B [12, 53, 54]). An auxiliary replay mechanism is introduced [55, 56] to maintain general-domain understanding for broader applications [33, 57]; a portion of CLIP training data (LAION-2B) is interleaved simultaneously with species contrastive learning. We ablate this decision and find that the experience replay improves biological understanding and performance across diverse tasks in §6.

We train BIOCLIP 2 on 32 NVIDIA H100 GPUs for 10 days on 214M organismal biology images with hierarchical labels and 26M randomly-sampled image-text pairs from LAION-2B for 30 epochs. We provide the training details in §D.

### 4.1 Species Classification Performance

We evaluate BIOCLIP 2 on species classification tasks in Table 1. We use the same benchmarks as BioCLIP [13], including seven tasks from Meta-Album [58] and Rare Species [59]. We substitute

Table 2: Biological visual tasks beyond species classification. **Bold** and underlined entries indicate the **best** and second best accuracies. See §H for task and evaluation methodology details.

| | Animals | | | Plants | | |
| --- | --- | --- | --- | --- | --- | --- |
| **Model** | FishNet | NeWT | AwA2 | Herb. 19 | PlantDoc | Mean |
| CLIP (ViT-L/14) | $27.9_{\pm0.2}$ | $83.4_{\pm0.1}$ | $61.6_{\pm0.6}$ | $18.2_{\pm0.1}$ | $22.3_{\pm3.3}$ | 42.7 |
| SigLIP | $31.9_{\pm0.1}$ | $83.2_{\pm0.1}$ | $\underline{67.3}_{\pm0.6}$ | $18.6_{\pm0.2}$ | $28.2_{\pm5.3}$ | 45.8 |
| Supervised-IN21K | $29.4_{\pm0.1}$ | $75.8_{\pm0.2}$ | $52.7_{\pm1.6}$ | $14.9_{\pm0.1}$ | $25.1_{\pm1.1}$ | 39.6 |
| DINOv3 | $\underline{37.9}_{\pm0.1}$ | $\underline{85.7}_{\pm0.0}$ | $48.0_{\pm2.8}$ | $\underline{31.2}_{\pm0.2}$ | $\underline{40.3}_{\pm1.2}$ | 48.6 |
| BioTrove-CLIP | $22.1_{\pm0.0}$ | $82.5_{\pm0.1}$ | $45.7_{\pm0.7}$ | $20.4_{\pm0.2}$ | $37.7_{\pm1.2}$ | 41.7 |
| BIOCLIP | $30.1_{\pm0.2}$ | $82.7_{\pm0.1}$ | $65.9_{\pm0.3}$ | $26.8_{\pm0.4}$ | $39.5_{\pm2.3}$ | $\underline{49.0}$ |
| BIOCLIP 2 | $\mathbf{39.8}_{\pm0.4}$ | $\mathbf{89.1}_{\pm0.1}$ | $\mathbf{69.5}_{\pm1.1}$ | $\mathbf{48.6}_{\pm0.6}$ | $\mathbf{40.4}_{\pm3.7}$ | $\mathbf{57.5}$ |

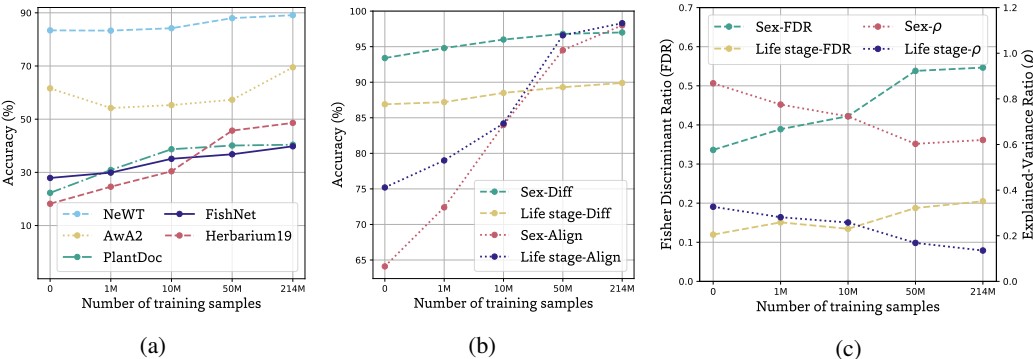

Figure 3: **(a)** The model performance on five downstream tasks under different scales of training data. **(b)** The model performance on differentiating and aligning different life stages and sexes. **(c)** The separation and orthogonality evaluation of models trained with different amounts of data.

Birds-525 [60] with NABirds [61] due to data inaccessibility. Additionally, we collect a test set of IDLE-OO Camera Traps from the Labeled Information Library of Alexandria: Biology and Conservation (LILA-BC) [62, 63, 64, 65, 66, 67] to illustrate more realistic species classification applications in the wild. The results suggest substantial improvements of BIOCLIP 2 over BIOCLIP. Particularly, attributed to more comprehensive species and image type coverage of TREEOFLIFE-200M, we observe over $20\%$ zero-shot improvement on Camera Trap, Fungi, and Rare Species. On average, BIOCLIP 2 surpasses the second-best model by $16.1\%$ and provides a $30.1\%$ improvement over the original CLIP model that serves as weight initialization. Information on baselines is in §E.

## 5 Emergent Properties from Scaling Hierarchical Contrastive Learning

### 5.1 Beyond Species Classification

Biology's organization extends beyond species taxonomies; if scaling truly induces emergent behavior, model representations learned through species-level supervision should transfer to problems far removed from species classification. We collect and benchmark models on five visual benchmarks that push past species ID: habitat classification (ecological context) [16], trait prediction (evolutionary studies) [18, 17], new-species identification (biodiversity monitoring) [19], and agricultural disease detection [20]. For each task, we keep the evaluated models frozen and extract the corresponding sample embeddings. The embeddings are subsequently processed using machine-learning techniques (*e.g.*, support vector classifiers). Detailed evaluation procedures are listed in §H.

Table 2 presents the performance comparison among BIOCLIP 2, vision-language baselines, and vision-only models. Although no information on these tasks is explicitly described during training, BIOCLIP 2 yields an average performance improvement of $14.8\%$ over the original CLIP baseline. DINOv3 is commonly believed to capture fine-grained visual features and is adopted for diverse visual tasks [22, 68, 69]. Nevertheless, BIOCLIP 2 yields an $8.9\%$ performance gap over DINOv3.

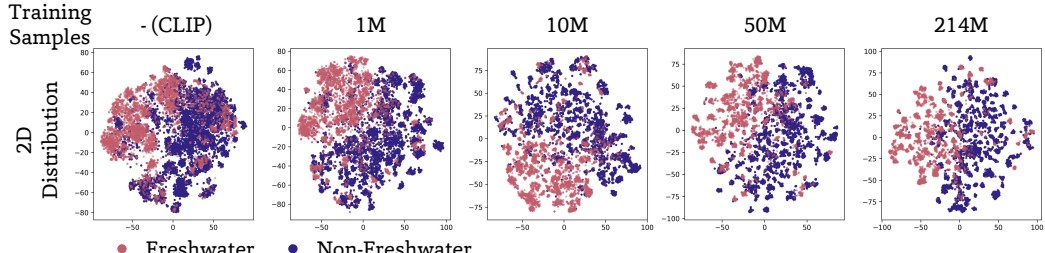

Figure 4: t-SNE embedding visualization of FishNet test set for models trained with different amounts of data. The leftmost plot is the original LAION-2B CLIP ViT-L/14. As the training data scales, freshwater fish become more distinct from saltwater fish and brackish fish, despite no explicit supervision, demonstrating that data scale contributes to emergent properties in model representations.

## 5.2 Scaling Trends

Better species classification with more species-labeled data is expected, but its effect on other, non-species classification tasks is unexpected. To better understand the relationship between training data scale and non-species classification performance, we apply the same hierarchical training with varying sizes of data: 1M, 10M, 50M, and 214M samples. The smaller datasets are randomly sampled from the complete set without losing taxa representativeness. We compare the performance of the baseline CLIP ViT-L/14 and the four obtained models on five non-species classification tasks in Figure 3a. A consistent improvement is observed as the volume of training data increases from 1M to 214M. In AwA2, for example, the 1M model is worse than the baseline. However, the model gradually learns more generalized representations for different attributes across species and obtains improved performance as data scales up.

Next, we investigate how scale affects representations within species. We collect two groups of images with intra-species appearance variations: life stage variations from NeWT [18] and sex variations from NABirds [61]. We ask whether scaling hierarchical contrastive training collapses all images of one species onto a single prototype or still distinguishes juveniles from adults and males from females. We accordingly design two complementary tasks for each type of variation: (i) *alignment*, where a species classifier trained on one variant (*e.g.*, juvenile images) is expected to recognize the species on the other variant (*e.g.*, adult images), and (ii) *differentiation*, where the task is to tell the variants apart (*e.g.*, juvenile *vs.* adult). As illustrated in Figure 3b, data scale steadily improves cross-variant species recognition. But at the same time, the model also becomes *better* at distinguishing the variants themselves.

## 5.3 Emergent Properties and Qualitative Analysis

Why does scaling data boost tasks that are never supervised during hierarchical contrastive training? We look deeper into BIOCLIP 2's embedding space and identify two emergent properties that generalize beyond species classification.

First, the embedding distribution of different species *aligns with their ecological and functional relationships*. Figure 4 shows t-SNE plots [70] of FishNet test set embeddings at four training scales, colored by whether the fish can live in freshwater or not. In the baseline CLIP plot (left), freshwater and non-freshwater fish have a large portion of overlap. Larger training sets progressively separate the two groups in the embedding space. We note that there is no explicit constraint to arrange meaningful distribution across species in contrastive loss, highlighting the emergence at scale.

Second, the intra-species variations are *preserved and separated*. Figure 5 shows t-SNE plots of BIOCLIP 2 embeddings of three species from NeWT exhibiting life-stage variation. In the 2D plots (top), different species (shown in different colors) tighten progressively from left to right, which is a direct consequence of scaled contrastive training. At the same time, the intra-species variations are preserved and better separated than the baseline CLIP model (leftmost sub-figure). We further project the embeddings onto 3D spaces created by singular value decomposition (SVD, bottom), which reveals that the intra-species variations lie in subspaces roughly orthogonal to the species span. Therefore, the existence of intra-species variation does not interfere with inter-species distinctions. §F contains more empirical observations.

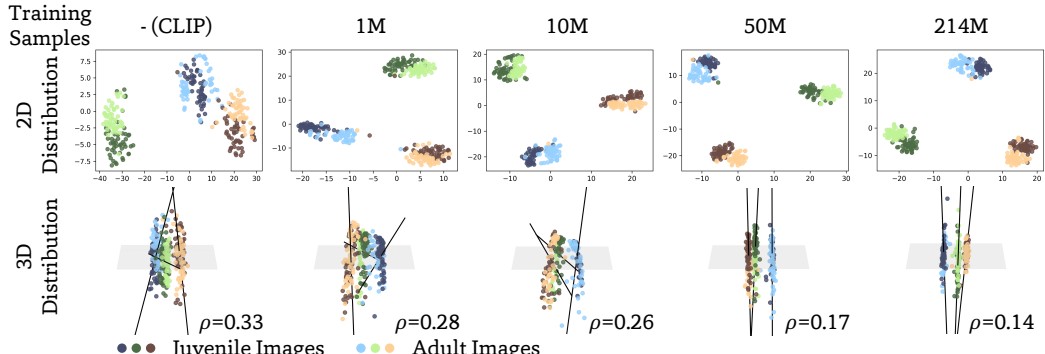

Figure 5: The embedding distribution of life stage variations under different scales of training data. The 2D distributions are obtained using t-SNE. For the 3D distributions, we first run SVD with the mean embedding of each species. The first two singular vectors are used to construct the gray plane that captures most inter-species differences. The embeddings are then projected into the 3D space with an additional orthogonal dimension. The straight lines point from the mean embedding of juvenile images to that of the adult images. As the training scales up, the intra-species variations are preserved in the subspace orthogonal to the inter-species differences. Orthogonality improves with data scale, as evidenced by the decreasing explained-variance ratio $\rho$.

## 5.4 Formal Analysis

We next ask *why* these two properties of inter-species ecological alignment and intra-species variation separation emerge with scale.

**Inter-species ecological alignment.** As training data increases, species that share proximal taxonomic labels are pulled toward common textual prototypes at multiple levels. As related taxa typically share morphology, behavior, and ecological characteristics, this multi-level supervision aligns visual similarity with functional similarity [15]. With more samples per species providing supervision at scale, embeddings of species in the same family or genus form coherent macro-clusters. In effect, the embeddings extracted by BIOCLIP 2 are more separable for different ecological groups.

**Intra-species variation separation.** While the inter-species ecological alignment can be explained by hierarchical supervision, the intra-species variations are *not* encoded in taxonomic labels. The preservation of intra-species variations also contradicts the common intuition of contrastive training effects. Therefore, we investigate the optimization of contrastive loss. We propose the following theorem to suggest that subspace orthogonal to inter-species differences is allowed after extensive training to accommodate intra-species variations.

**Theorem 5.1.** *Let $\boldsymbol{\mu}$ be the prototypes of species, with $\boldsymbol{\mu}_s$ as the prototype of species $s$. Let $\tau$ be temperature. If different $\boldsymbol{\mu}_k$ are nearly orthogonal (*i.e.*, species are well separated), the intra-species variation $\delta$ for species $s$ is constrained by*

$$\boldsymbol{\delta}^\top \left[ \frac{1}{2\tau^2} \left( \sum_k w_k \boldsymbol{\mu}_k \boldsymbol{\mu}_k^\top - \boldsymbol{\mu}_s \boldsymbol{\mu}_s^\top \right) \right] \boldsymbol{\delta}, \quad where \quad w_k = \frac{\exp(\boldsymbol{\mu}_s^\top \boldsymbol{\mu}_k)/\tau}{\sum_k \left( \exp(\boldsymbol{\mu}_s^\top \boldsymbol{\mu}_k)/\tau \right)}.$$

*Proof.* See §C. □

Thus, as long as the variation $\boldsymbol{\delta}$ is distributed in a subspace orthogonal to the inter-species distinctions, the scale of $\boldsymbol{\delta}$ won't interfere with the overall contrastive optimization. The orthogonality is qualitatively supported by Figure 5. To further quantify it, we calculate the explained-variance ratio, *i.e.*, the ratio in the intra-species variation that is captured by the species span [71]. We first obtain an orthonormal basis for the species prototypes $\boldsymbol{U}$ using QR decomposition. Let $\boldsymbol{D} = [\boldsymbol{d}_1, \ldots, \boldsymbol{d}_n] \in \mathbb{R}^{d \times n}$ be the matrix stacking $n$ intra-species variation difference vectors with dimension $d$. The explained-variance ratio calculates the energy fraction inside species span by $\rho = \|\boldsymbol{U}^\top \boldsymbol{D}\|_F^2 / \|\boldsymbol{D}\|_F^2$, where $\|\cdot\|_F$ denotes the Frobenius norm [72]. We show the ratio change as the data scales up in Figure 3c. The results suggest that the intra-species variations are increasingly orthogonal to the species differences. Due to the orthogonality, the existence of intra-species variations will not interfere with the inter-species distinctions. The observation of smaller projection areas in Figure 5 at the species span also indicates better species classification accuracy after extensive contrastive training.

Table 3: Ablation study with different training settings. BIOCLIP 2 adopts hierarchical contrastive training with taxonomic labels. We ablate using scientific names solely without taxonomic labels, and one-hot labels with cross-entropy loss instead of the contrastive objective.

| Dataset | Hierarchical Contrastive | Contrastive w/ Scientific Name | Cross-entropy Loss |
|---|---|---|---|
| FishNet | $\mathbf{35.1}_{\pm 0.1}$ | $33.8_{\pm 0.1}$ | $33.0_{\pm 0.1}$ |
| PlantDoc | $\mathbf{38.7}_{\pm 3.7}$ | $37.3_{\pm 3.3}$ | $30.9_{\pm 1.9}$ |
| Life stage-Diff | $88.0_{\pm 0.1}$ | $\mathbf{88.5}_{\pm 0.2}$ | $85.5_{\pm 0.2}$ |
| Life stage-Align | $84.1_{\pm 0.1}$ | $\mathbf{84.5}_{\pm 0.1}$ | $78.6_{\pm 0.1}$ |
| Sex-Diff | $\mathbf{97.0}_{\pm 0.1}$ | $96.6_{\pm 0.1}$ | $95.5_{\pm 0.1}$ |
| Sex-Align | $\mathbf{84.1}_{\pm 0.2}$ | $82.7_{\pm 0.3}$ | $74.9_{\pm 0.2}$ |

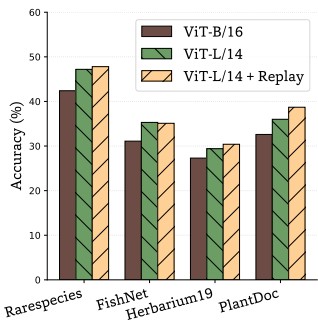

Figure 6: Ablation study on model size and experience replay.

Furthermore, we look at the separation between these variations using the Fisher Discriminant Ratio (FDR) metric [73]. Given two variant classes $A$ and $B$, FDR is defined by their embedding mean $\boldsymbol{\mu}$ and standard deviation $\sigma$: FDR $= \|\boldsymbol{\mu}_A - \boldsymbol{\mu}_B\|^2/(\sigma_A^2 + \sigma_B^2)$. The increasing trends in Figure 3c support that, in addition to orthogonality, the intra-species variations are more separable as the training scales up. This trend is also qualitatively evidenced by the top row of Figure 5. These theoretical and empirical insights validate that BIOCLIP 2 learns to preserve and separate intra-species variations without explicit training constraints. We provide more qualitative analyses in §F.2.

## 6 Ablation Study and Analysis

**The necessity of contrastive loss and taxonomic labels.** The previous analyses highlight the effectiveness of hierarchical contrastive training. Table 3 ablates two key modeling decisions using TREEOFLIFE-10M as a feasible testbed: scientific names solely vs. hierarchical labels and contrastive learning vs. cross-entropy loss on one-hot labels. The experiments are conducted on TREEOFLIFE-10M data with ViT-L/14 as the visual encoder. When trained solely with scientific names, the model loses some of the hierarchical supervision embedded in taxonomic labels. As a result, the performance on FishNet drops by $1.3\%$. However, contrastive training still leads to separation for both species and intra-species variations. Training a 952K-class softmax classifier with cross-entropy loss is optimization-heavy and leads to inferior performance on all benchmarks. The adopted hierarchical contrastive supervision leverages the advantages of both aspects and yields the best overall performance across benchmarks.

**Architecture and replay.** BIOCLIP 2 scales up the visual encoder of BIOCLIP from a ViT-B/16 to ViT-L/14 and introduces experience replay of CLIP training data (LAION-2B). We ablate the effects of these two changes, again using TREEOFLIFE-10M as a testbed. Figure 6 shows that increasing model capacity improves performance across all benchmarks. Comparatively, experience replay leads to better species classification accuracy and improved performance on some of the other visual tasks. We provide a more detailed empirical study of experience replay in §F.1.

## 7 Conclusion

In this work, we curate TREEOFLIFE-200M, the largest and most diverse biological organism dataset to date, and train BIOCLIP 2 with hierarchical taxonomic labels. BIOCLIP 2 achieves state-of-the-art accuracy on species classification. More importantly, large-scale training gives rise to two emergent properties not described during training. At the inter-species level, the embedding distribution of different species aligns with their ecological relationships. At the intra-species level, the appearance variations within species are preserved and well separated in the embedding space. We demonstrate that combining the effort of domain-specific scaling and structured supervision leads to effective generalization beyond the initial training objectives. BIOCLIP 2 serves as a strong foundation model for biological research and simultaneously evidences the effectiveness of scale-driven scientific discovery.

## Acknowledgments and Disclosure of Funding

We would like to thank Zhiyuan Tao, Shuheng Wang, Ziheng Zhang, Zhongwei Wang, and Leanna House for their help with the TREEOFLIFE-200M dataset, Charles (Chuck) Stewart, Sara Beery, and other Imageomics Team members for their constructive feedback and Sergiu Sanielevici, Tom Maiden, and TJ Olesky for their dedicated assistance with arranging the necessary computational resources.

We are grateful to Kakani Katija and Dirk Steinke for helpful conversations regarding use and integration of FathomNet and BIOSCAN-5M, respectively, as well as Stephen Formel and Markus Döring for GBIF. We thank Marie Grosjean for comparative methods for filtering citizen science images and Dylan Verheul for assistance with acquiring images from observation.org from GBIF. We thank Suren Byna for a helpful conversation on early dataset design decisions. We thank Doug Johnson for his collaboration in hosting this large dataset on the Ohio Supercomputer Center research storage file system.

Our research is supported by NSF OAC 2118240 and resources from the Ohio Supercomputer Center [74]. This work used the Bridges-2 system, which is supported by NSF award number OAC-1928147 at the Pittsburgh Supercomputing Center (PSC) [75], under the auspices of the NAIRR Pilot program.

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

## Appendix

The Appendix is organized as follows:

- In §A we discuss limitations of our work.
- In §B we discuss the broader impacts of our work.
- In §C we present the proof of Theorem 5.1.
- In §D we describe the training implementation for BIOCLIP 2 in detail.
- In §E we introduce the baselines employed for the numerical experiments.
- In §F we demonstrate more empirical observations of BIOCLIP 2.
- In §G we discuss the relationship between the adopted training scheme and neural collapse.
- In §H we describe both the biological visual benchmarks and the implementation details.
- In §I we provide the details of data processing for TREEOFLIFE-200M.
- In §J we list the contribution of each author.

## A  Limitations

**Theoretical limitation.** In this work, we have proved that the intra-species variations are preserved in subspaces orthogonal to the inter-species difference. We also have empirical observations that the variations are more separable as the training data scales up (See Figure 3c). However, we haven't formally proved this. It will be our future work to have deeper theoretical analyses of the separation of intra-species variations to better understand BIOCLIP 2's emergent properties.

**Data limitation.** TREEOFLIFE-200M is an imbalanced dataset in both taxonomic coverage and image type. Specifically, the dataset exhibits a long-tailed distribution across taxa. This is to be expected when working with biological data—not all taxonomic ranks are represented evenly across the tree of life. For instance, though TREEOFLIFE-200M has a balanced representation (at the kingdom level) between plants and animals, animals represent a larger proportion of described species [51].

Some of this is due to the nature of the image type distribution, which we provide for GBIF (Camera-trap, Citizen Science, and eleven Museum Specimen types: Fungi, Insect, Invertebrate Zoology, Microbiology, Plant, Vertebrate Zoology - [Amphibians, Birds, Fishes, Mammals, Others], as well as Unidentified). EOL contains the same categories, but we do not have precise numbers; BIOSCAN-5M are essentially all insect museum specimens, though the images are taken by researchers, so will skew toward their area of study; FathomNet contains a mix. Citizen Science images are the vast majority (151M from GBIF alone); these will skew toward more charismatic species and plants. Our next largest category is museum specimen images (51.8M from GBIF), which are limited more to representatives of the species, so may not contain as many images per taxa, though more taxa are represented. Finally, camera trap images make up the smallest portion of the dataset (617.8K in GBIF), and when filtered, these are only images of animals and generally those large enough to trigger a motion sensor or be detected in the primary provider's post-processing.

Further emphasis on the impact of citizen science images in amassing larger representations of species: the most prevalent taxonomic classes, flowering plants, insects, birds, mushrooms, and mammals, have millions of representative images, while the least-represented, microscopic organisms (e.g., bacteria, viruses), have a dozen or fewer.

## B  Broader Impacts

BIOCLIP 2 and TREEOFLIFE-200M provide great potential to improve and enhance existing conservation efforts, in particular by facilitating recognition of threatened species. As noted in §3.3, TREEOFLIFE-200M has expansive coverage of threatened species, as classified by IUCN. It additionally builds on the coverage of species considered to be Data Deficient. Based on the emergent properties displayed by BIOCLIP 2, there is potential to add to the effort to understand the risks facing these species that cannot currently be classified by IUCN due to lack of available information, as suggested in [76]. These designations are crucial to the international effort to protect biodiversity across the planet.

Unfortunately, as with many open-source efforts to further conservation goals, there is potential for bad actors to make use of these tools for malicious purposes. Though the improvement on threatened species *could* make it easier for poachers to identify protected species, these types of tools are a force-multiplier to monitor illicit trade and sales of these same species. The primary risk to endangered species comes from disclosure of precise location information rather than improved classification capability [77]. Our data does not provide geo-tagged information on the organisms included, minimizing the vulnerabilities that could be used in poaching.

## C    Proof of Theorem 5.1

*Proof.* The contrastive loss for one visual embedding $z$ belonging to the class $s$ and the corresponding text embedding $c_s$ is:

$$l(z, c_s) = -\log \frac{\exp(z^\top c_s/\tau)}{\sum_k \exp(z^\top c_k/\tau)}. \tag{1}$$

Let $h_\phi(L(\cdot))$ be the text encoder. Assume the representation is already close to the species prototype $\mu_s = h_\phi(L(s)) = c_s$, and there is a residual $\delta$ representing the intra-species variance:

$$\delta := z - \mu_s, \quad \|\delta\| \ll \|\mu_s - \mu_{k \neq s}\|.$$

Define:

$$a_k := \frac{\mu_s^\top \mu_k}{\tau}, \quad Z = \sum_k \exp(a_k), \quad w_k := \frac{\exp(a_k)}{Z}$$

Substituting $z = \mu_s + \delta$ into Equation 1:

$$l(\mu_s + \delta, c_s) = -\frac{(\mu_s + \delta)^\top \mu_s}{\tau} + \log\left[\sum_k \exp\left(\frac{(\mu_s + \delta)^\top \mu_k}{\tau}\right)\right] \tag{2}$$

$$= -\frac{\mu_s^\top (\mu_s + \delta)}{\tau} + \log\left[\sum_k \exp\left(\frac{\mu_s^\top \mu_k}{\tau}\right) \cdot \exp\left(\frac{\delta^\top \mu_k}{\tau}\right)\right] \tag{3}$$

$$= -\frac{\mu_s^\top \mu_s}{\tau} - \frac{\mu_s^\top \delta}{\tau} + \log\left[\sum_k \exp\left(\frac{\mu_s^\top \mu_k}{\tau}\right) \sum_k \frac{\exp(\mu_s^\top \mu_k/\tau)}{\sum_k \exp(\mu_s^\top \mu_k/\tau)} \cdot \exp\left(\frac{\delta^\top \mu_k}{\tau}\right)\right] \tag{4}$$

$$= -\frac{\mu_s^\top \mu_s}{\tau} - \frac{\mu_s^\top \delta}{\tau} + \log Z + \log\left[\sum_k w_k \exp\left(\frac{\delta^\top \mu_k}{\tau}\right)\right]. \tag{5}$$

Use the Taylor expansion to the second order for the argument of the logarithm of the last term $\Psi(\delta) = \log\left[\sum_k w_k \exp(\delta^\top \mu_k/\tau)\right]$:

$$\sum_k w_k \exp\left(\frac{\delta^\top \mu_k}{\tau}\right) = 1 + \sum_k w_k \frac{\delta^\top \mu_k}{\tau} + \frac{1}{2}\sum_k w_k \frac{(\delta^\top \mu_k)^2}{\tau^2} + O(\|\delta\|^3), \tag{6}$$

$$\text{obtaining} \quad \Psi(\delta) = \sum_k w_k \frac{\delta^\top \mu_k}{\tau} + \frac{1}{2}\left[\sum_k w_k \frac{(\delta^\top \mu_k)^2}{\tau^2} - \left(\sum_k w_k \frac{\delta^\top \mu_k}{\tau}\right)^2\right] + O(\|\delta\|^3). \tag{7}$$

Insert $\Psi(\delta)$ back into Eq.2:

$$l = l(\mu_s, \mu_s) - \frac{\mu_s^\top \delta}{\tau} + \sum_k w_k \frac{\delta^\top \mu_k}{\tau}$$

$$+ \frac{1}{2}\left[\sum_k w_k \frac{(\delta^\top \mu_k)^2}{\tau^2} - \left(\sum_k w_k \frac{\delta^\top \mu_k}{\tau}\right)^2\right] + O(\|\delta\|^3).$$

Define $\boldsymbol{m} := \sum_k w_k \boldsymbol{\mu}_k$. Then collecting first-order terms around $\boldsymbol{\delta}$ results in the following:

$$\frac{(\sum_k w_k \boldsymbol{\mu}_k - \boldsymbol{\mu}_s)^\top \boldsymbol{\delta}}{\tau} = \frac{(\boldsymbol{m} - \boldsymbol{\mu}_s)^\top \boldsymbol{\delta}}{\tau}.$$

Suppose the training resulted in $w_s \approx 1$, leading to $\boldsymbol{m} \approx \boldsymbol{\mu}_s$. Then the first-order terms will vanish as the embeddings of different species are better separated.

We further rewrite the second-order term of Eq.6 as:

$$\frac{1}{2}\left[\sum_k w_k \frac{(\boldsymbol{\delta}^\top \boldsymbol{\mu}_k)^2}{\tau^2} - \left(\sum_k w_k \frac{\boldsymbol{\delta}^\top \boldsymbol{\mu}_k}{\tau}\right)^2\right] = \frac{1}{2}\left[\sum_k w_k \frac{(\boldsymbol{\delta}^\top \boldsymbol{\mu}_k)^2}{\tau^2} - \frac{(\boldsymbol{\delta}^\top \boldsymbol{m})^2}{\tau^2}\right]$$

$$= \boldsymbol{\delta}^\top \left[\frac{1}{2\tau^2}\left(\sum_k w_k \boldsymbol{\mu}_k \boldsymbol{\mu}_k^\top - \boldsymbol{m}\boldsymbol{m}^\top\right)\right] \boldsymbol{\delta}.$$

Substituting $\boldsymbol{m} \approx \boldsymbol{\mu}_s$, we have an approximation as

$$\boldsymbol{\delta}^\top \left[\frac{1}{2\tau^2}\left(\sum_k w_k \boldsymbol{\mu}_k \boldsymbol{\mu}_k^\top - \boldsymbol{\mu}_s \boldsymbol{\mu}_s^\top\right)\right] \boldsymbol{\delta}.$$

The Hessian matrix lies in the span of species, as each term is an outer product of a prototype. Therefore, as long as the residual $\boldsymbol{\delta}$ is distributed in a subspace orthogonal to the species differences, the scale of $\boldsymbol{\delta}$ will not interfere with the overall contrastive optimization. $\qquad\square$

## D    Training Implementation Details

Table 4: The adopted hyper-parameter setting in training BIOCLIP 2.

| Hyper-parameter | Value |
|---|---|
| Architecture | ViT-L/14 |
| Optimizer | Adam |
| Batch size/GPU (organism) | 2,816 |
| Batch size/GPU (replay) | 320 |
| GPUs | 32 H100s |
| Epochs | 30 |
| Max learning rate | $1 \times 10^{-4}$ |
| Warm-up steps | 1,875 |
| Weight decay | 0.2 |
| Input resolution | 224 |

Table 5: The adopted hyper-parameter setting in the ablation study.

| Hyper-parameter | Value |
|---|---|
| Architecture | ViT-L/14 |
| Optimizer | Adam |
| Batch size/GPU (organism) | 2,816 |
| Batch size/GPU (replay) | 320 |
| GPUs | 8 H100s |
| Epochs | 100 |
| Max learning rate | $1 \times 10^{-4}$ |
| Warm-up steps | 1,500 |
| Weight decay | 0.2 |
| Input resolution | 224 |

We list the adopted hyperparameters in Table 4 and Table 5 for the BIOCLIP 2 training and ablation study, respectively. TREEOFLIFE-10M is used for the ablation study. The batch size presented in both tables is the size per GPU. In addition to the larger GPU number and smaller batch size compared with the training in BIOCLIP due to the larger model size, another important modification is the introduction of experience replay. An additional visual projector is introduced as described in §F.1. Beyond the visual projector, the model architecture is kept the same as that of CLIP [12]. During the evaluation, all the embeddings are extracted with the visual projector for hierarchical label matching to avoid extra influence.

## E    Baseline Details

In the quantitative experiments, we compare BIOCLIP 2 with the following baseline vision-language models and vision-only models. Without specification, the input image size is 224.

- **CLIP (ViT-L/14).** We compare BIOCLIP 2 with the CLIP model pre-trained on LAION-2B [54], which has the same architecture and patch size. The CLIP model is also used as the weight initialization of the BIOCLIP 2 training. It uses a ViT-large visual encoder with a patch size of 14. We load the weight from the OpenCLIP repository.

Table 6: Performance comparison between different replay designs of CLIP training data. All the models in the bottom three rows are initialized with CLIP (the first row) and trained with TREEOFLIFE-10M data. The CLIP model is pre-trained with LAION-2B data, from which we randomly select 2M samples for this experiment. $\Delta$ represents the performance gap over the CLIP baseline. **Bold** entries indicate the best accuracy.

| | Species | | Non-Species | | | | | | | |
| Model | NABirds | Rarespecies | FishNet | NeWT | AWA2 | Herb. 19 | PlantDoc | INQUIRE | Mean ($\Delta$) | |
|---|---|---|---|---|---|---|---|---|---|---|
| CLIP (ViT-L/14) | 66.5 | 35.2 | $27.9_{\pm0.2}$ | $83.4_{\pm0.1}$ | $61.6_{\pm0.6}$ | $18.2_{\pm0.1}$ | $22.3_{\pm3.3}$ | 35.0 | 43.8 | – |
| No Replay | 68.9 | 46.1 | $\mathbf{35.3}_{\pm0.1}$ | $83.8_{\pm0.1}$ | $58.0_{\pm2.8}$ | $29.4_{\pm0.4}$ | $36.0_{\pm2.8}$ | 34.4 | 49.0 | ↑5.2 |
| Single-proj | 68.8 | 44.8 | $34.4_{\pm0.2}$ | $\mathbf{84.5}_{\pm0.2}$ | $\mathbf{58.1}_{\pm1.4}$ | $\mathbf{31.0}_{\pm0.2}$ | $38.1_{\pm2.4}$ | 34.7 | 49.3 | ↑5.5 |
| Separate-proj (Ours) | **71.2** | **47.2** | $35.1_{\pm0.1}$ | $84.2_{\pm0.1}$ | $57.4_{\pm0.9}$ | $30.4_{\pm0.3}$ | $\mathbf{38.7}_{\pm3.7}$ | **37.1** | **50.2** | ↑6.4 |

- **SigLIP.** In addition to the standard CLIP model, we also evaluate the performance of SigLIP [21]. We adopt the SigLIP model pre-trained on WebLI data [78]. The adopted visual encoder is ViT-large [53], the patch size is 16, and the input image resolution is 256. The model weight is also loaded from the OpenCLIP repository.
- **Supervised-IN21K.** For vision-only models, we first select a ViT-large model [53] trained in a supervised way on ImageNet-21K dataset [79]. As it is a vision-only model, we only run a few-shot and non-species classification tasks with it. The patch size is 32. The model is publicly downloadable in Hugging Face.
- **DINOv3.** Besides supervised pre-training, we also evaluate the performance of DINOv3, which is pre-trained in an unsupervised way [22]. The backbone architecture is ViT-large, and the patch size is 16. The model can be downloaded from Hugging Face. Similarly, we only run few-shot and non-species classification evaluations with DINOv3.
- **BioTrove-CLIP.** The above four models are trained on general knowledge covering a variety of topics. We also compare BIOCLIP 2 with domain-specific models. BioTrove-CLIP is trained with BioTrove [14]. The model weights can be downloaded from Hugging Face. Among the provided three models, we use the model initialized with OpenAI CLIP [12] that yields the best average accuracy [14]. The visual backbone is ViT-base with a patch size of 16.
- **BIOCLIP.** BIOCLIP is trained on TREEOFLIFE-10M. It adopts ViT-base as the visual encoder, with a patch size of 16. The model weight is publicly available in Hugging Face.

# F More Empirical Observations

In this section, we present more empirical observations of the training design, detailed numerical results, and qualitative analyses.

## F.1 CLIP Training Data Replay

Together with the contrastive supervision of hierarchical labels, we also introduce the replay of CLIP training data to retain the understanding of general knowledge. For each experiment of different training scales, we randomly select a subset from LAION-2B with $10\%$-$20\%$ of the corresponding biological image total. Specifically, for the largest run on 214M biological images, we select 26M samples from LAION-2B. Each training batch consists of 69,312 biological images and 8,192 replay samples. However, the text labels in TREEOFLIFE-200M are primarily different forms of taxonomic names, which have a distributional gap from the CLIP training data. Therefore, we apply a separate visual projector specifically for the replay data to avoid the optimization conflict. Other than this difference, the biological data and replay data share the same visual backbone and text encoder.

We evaluate different replay settings quantitatively in Table 6. The "No Replay" row shows the baseline performance applying biological contrastive training on top of the pre-trained CLIP model, where a $5.2\%$ performance improvement is achieved. When the replay data is added, which shares the visual projector with biological images, we observe a conflicting performance change. On Herb. 19 [19] and PlantDoc [20], more than $2\%$ improvement is acquired. However, the species classification accuracy on Rare Species drops by $1.3\%$. We attribute the inconsistent performance change to the distribution gap between biological and replay text labels. After applying a separate

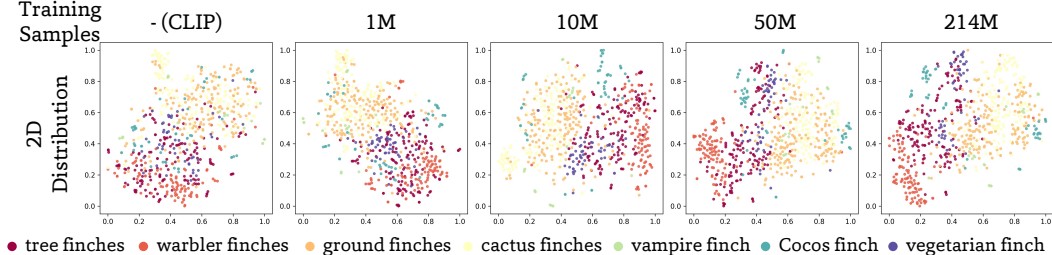

Figure 7: The t-SNE distribution of Darwin's finches under different scales of training data. Different colors represent different groups of finch species. As the training data scales up, the embeddings form biologically meaningful clusters that align with the phylogenetic tree and their functional traits.

visual projector for the replay data, we observe overall improved performance across multiple benchmarks. At the same time, we also notice that replay has limited influence on benchmarks like AwA2 [17]. Additionally, we evaluate the preservation of general knowledge understanding on INQUIRE-Rerank [57]. Applying contrastive training with taxonomic labels slightly hurts the performance, while the single-projector replay fails to retain the understanding. Comparatively, the adopted separate-projector replay improves the performance by 2.1%. Given that there is still a distribution gap between the taxonomic labels and natural language, we do not treat INQUIRE as one of our main focuses in this work.

## F.2 Qualitative Analyses

**Embedding distribution of Darwin's finches.** In Figure 1, we show that the embedding distribution of Darwin's finches aligns with the beak size. Here we further visualize the distribution under different training scales in Figure 7. Based on the genome-based phylogeny, warbler finches were the most ancient branches, while tree finches and ground finches form the recent branches [23]. Among these species, warbler finches have the smallest beak, convenient for extracting tiny arthropods from leaves. Comparatively, ground finches have larger beaks, which are more suitable for cracking seeds and nuts. In the original CLIP embedding space, the warbler finches and tree finches are mixed. As the training data scales up, these two groups are separated, and the relative geometric relationship of all the finches aligns with their phylogeny tree. While the species separation is induced by taxonomic supervision, BIOCLIP 2's embedding space again illustrates emergent higher-level biological meaningfulness after extensive training.

**Embedding distribution of Sex data.** Similar to Figure 5, we visualize the 2D and 3D distributions of embeddings from 3 species of the Sex data in Figure 8. We can draw conclusions similar to those obtained from Figure 5. When no training data is incorporated, the embeddings extracted by the original CLIP visual encoder (leftmost sub-figure) demonstrate large portions of overlap between male and female images. After extensive vision-language contrastive training, the embeddings present clear decision boundaries between the two variants. Furthermore, as evidenced in the 3D distribution, the embeddings within each species form more compact clusters when projected onto the species span (the gray plane). Comparatively, instead of being eliminated after contrastive training, the intra-species variations are embedded in the subspace orthogonal to the inter-species differences. The extensive training facilitates BIOCLIP 2 to acquire a biologically meaningful embedding space, highlighting its value in serving as a biology foundation model.

**Embedding distribution of PlantDoc data.** We further visualize the distribution of 6 classes from the PlantDoc dataset in Figure 9. When no training is processed (leftmost sub-figure), embeddings of different species, as well as diseases, are mixed. As the training scale increases, we observe two trends. First, the margin between different species is enlarged, and the embeddings belonging to the same species are clustered together. Second, the diseased leaves are easier to separate within each species, although not explicitly constrained during training. More interestingly, the embeddings of healthy apple leaves are distributed close to the healthy blueberry leaves. These observations again highlight the biologically meaningful embedding space of BIOCLIP 2 after extensive training.

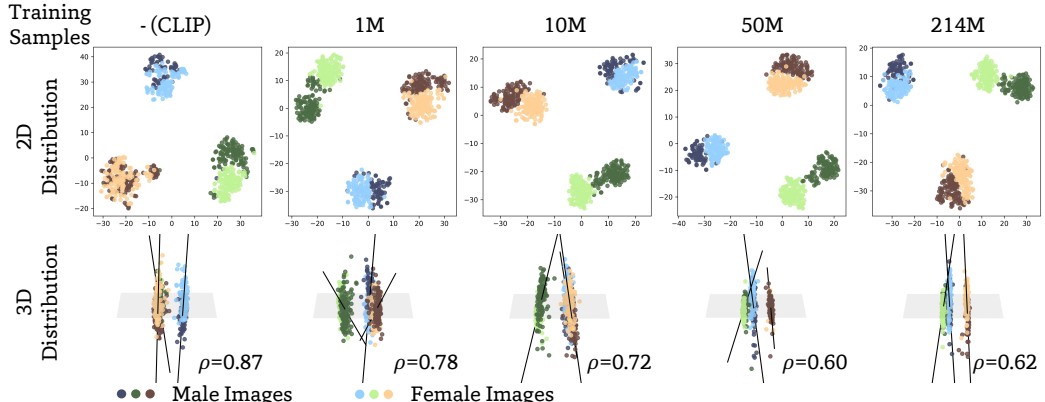

Figure 8: The embedding distribution of sex variations under different scales of training data. The 2D distributions are obtained using t-SNE. For the 3D distributions, we first run SVD with the mean embedding of each species. The first two singular vectors are used to construct the gray plane that captures most inter-species differences. The embeddings are then projected into the 3D space with an additional orthogonal dimension. The straight lines point from the mean embedding of male images to that of the female images. As the training data scales up, the intra-species variations are preserved in the subspace orthogonal to the inter-species differences. Orthogonality improves with data scale, as evidenced by the decreasing explained-variance ratio $\rho$.

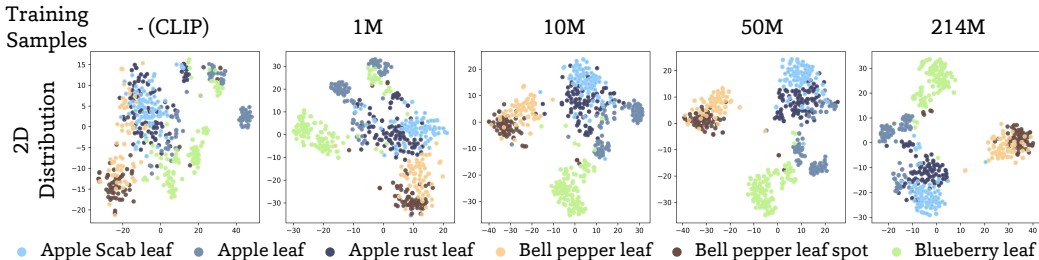

Figure 9: The t-SNE distribution of 6 classes from the PlantDoc dataset, including three species and three different diseases. As the training data scales up, not only are the species better separated, but the intra-species variations also form clusters, making them easier to separate.

### F.3 The decay of FDR numerator/denominator along the data scale.

We observe that FDR between two intra-species variation classes is increasing as the training data scales up. More specifically, we look into the numerator term (the difference between two class centers) and the denominator term (the variation of features). We visualize the curve of the numerator and denominator terms in Figure 10a and Figure 10b for life stage data and sex data, respectively. The y-axes are scaled to the same maximum ratio to the minimum value. As the number of training samples scales up from 1M to 214M, the denominator term goes through a larger decay than the numerator term. The numerical results further support the increasing FDR trend.

### F.4 Camera Trap Results

In addition to the species classification benchmarks adopted in [13], we further introduce a balanced camera trap image benchmark for species classification, IDLE-OO Camera Traps, derived from LILA-BC datasets [62], to construct a more realistic application scenario. Specifically, we select five datasets from LILA-BC that are labeled to the image level to avoid testing on noisy images—those labeled as containing an animal when it is simply the animal's habitat. The Island Conservation Camera Traps [63] were of particular interest for their stated purpose of assisting in the prevention of endangered island species' extinction and the varied ecosystems represented. This provides a fine-grained complement to the Rare Species test set [13, 59]. The Desert Lion Conservation Camera Traps dataset [64] is similarly intended to advance conservation efforts. With the Orinoquía Camera Traps [65], Ohio Small Animals [66], and ENA24 [67] camera trap datasets, we can test on camera

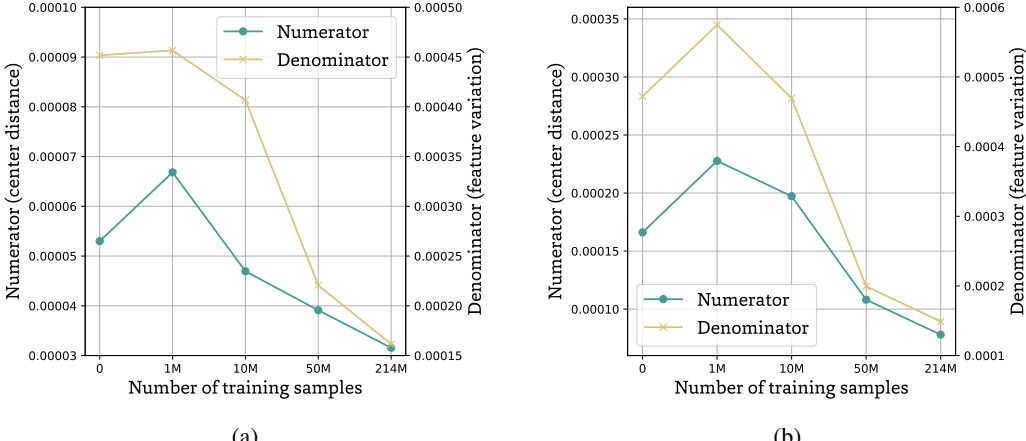

(a)                                        (b)

Figure 10: The decay curves for the numerator (difference between two class centers) and the denominator (feature variation) terms of the FDR metric along the increasing data scale. (a) The curves of life stage data. (b) The curves of sex data.

## G   Discussion on the Relationship with Neural Collapse

Neural collapse is a status where the intra-class variations collapse to zero, and the embeddings collapse to their corresponding class prototypes [80, 81]. The class prototypes collapse to the vertices of a simplex Equiangular Tight Frame (ETF). It has been empirically observed and theoretically proved that the commonly used loss functions—including cross-entropy loss and supervised contrastive loss [82]—lead to neural collapse at the terminal phase of training [24, 25, 83, 84]. If neural collapse happens, the intra-species appearance variations will be hardly separable. However, we reveal that after extensive training of BIOCLIP 2, the intra-species variations are preserved in the subspace orthogonal to inter-species differences, indicating that BIOCLIP 2 does not suffer neural collapse.

**Why BIOCLIP 2 does not lead to Neural Collapse.** We summarize two key reasons that allow the existence of the intra-species variation subspace in BIOCLIP 2.

First, the class prototypes in standard classification tasks and supervised contrastive learning (SCL) can both be treated as fully trainable parameters [82]. In standard classification, the prototypes are the weights of the linear classifier. In SCL, they are simply the mean embedding of the corresponding classes. The class prototypes will form a simplex ETF after extensive training. Comparatively, the adopted contrastive training scheme in BIOCLIP 2 employs text embeddings of the taxonomic labels as class prototypes. During the creation of taxonomic labels, hierarchical structures have been naturally embedded into them based on ecological and functional evidence. Different species can share higher taxonomic levels, which makes them hard negatives. Therefore, even if the text encoder is also being trained, the generated prototypes will not become ETF.

Second, TREEOFLIFE-200M poses an enormous label space, where 952K classes are involved in training. Typical SCL is performed upon CIFAR or ImageNet-level datasets, consisting of $100 - 1,000$ classes. Previous works analyzing neural collapse usually assume the dimension of embeddings is larger than the class number and the sample number is balanced across different classes [25, 81]. In contrast, the class number involved in BIOCLIP 2 is much larger than the embedding dimension (768 channels). These conditions restrict the possibility of prototypes forming ETF.

trap images across varied settings. The Island Conservation set uses common names, while the remaining datasets are reduced to just those that are labeled to the species level, and then sampled to create a balanced test set across the remaining classes (these are evaluated on scientific names). The sampling and image manifests are provided in IDLE-OO Camera Traps. We report the detailed accuracy on each camera trap dataset in Table 7.

Table 7: Zero-, one-, and five-shot species classification accuracy of IDLE-OO Camera Traps from the Labeled Information Library of Alexandria: Biology and Conservation (LILA-BC) [62, 63, 64, 65, 66, 67] for different models. **Bold** and underlined entries indicate the **best** and second best accuracies, respectively.

| Model | Desert Lion | ENA24 | Island | Orinoquia | OH Small Animals | Mean |
|---|---|---|---|---|---|---|
| | | | **LILA-BC** | | | |
| Random Guessing | 3.1 | 5.0 | 3.2 | 3.6 | 2.6 | 3.5 |
| *Zero-Shot Classification* | | | | | | |
| CLIP (ViT-L/14) | 35.2 | 38.2 | 27.1 | 25.6 | 21.6 | 29.5 |
| SigLIP | 46.9 | 41.0 | 30.0 | 31.9 | 20.5 | 34.1 |
| BioTrove-CLIP | 9.7 | 10.4 | 14.5 | 8.0 | 12.4 | 11.0 |
| BIOCLIP | 47.2 | 42.5 | 19.0 | 27.1 | 23.1 | 31.8 |
| BIOCLIP 2 | **58.8** | **68.5** | **42.3** | **47.9** | **51.5** | **53.8** |
| *One-Shot Classification* | | | | | | |
| CLIP (ViT-L/14) | $37.6_{\pm2.5}$ | $35.9_{\pm3.8}$ | $44.7_{\pm3.1}$ | $31.3_{\pm2.7}$ | $30.7_{\pm2.0}$ | 36.0 |
| SigLIP | $41.1_{\pm1.8}$ | $\underline{39.1}_{\pm4.0}$ | $48.5_{\pm2.9}$ | $32.7_{\pm1.6}$ | $27.6_{\pm2.5}$ | 37.8 |
| Supervised-IN21K | $32.4_{\pm2.0}$ | $28.4_{\pm3.2}$ | $40.1_{\pm3.5}$ | $26.3_{\pm2.4}$ | $26.0_{\pm2.4}$ | 30.6 |
| DINOv3 | $\underline{48.1}_{\pm1.3}$ | $38.5_{\pm3.2}$ | $\mathbf{52.3}_{\pm4.0}$ | $\underline{40.0}_{\pm2.6}$ | $\underline{36.1}_{\pm2.3}$ | 43.0 |
| BioTrove-CLIP | $30.5_{\pm1.7}$ | $30.5_{\pm2.2}$ | $37.9_{\pm2.1}$ | $29.0_{\pm2.6}$ | $28.3_{\pm3.1}$ | 31.2 |
| BIOCLIP | $39.9_{\pm2.2}$ | $34.4_{\pm2.7}$ | $45.7_{\pm3.0}$ | $27.5_{\pm3.7}$ | $27.5_{\pm2.2}$ | 35.0 |
| BIOCLIP 2 | $\mathbf{54.3}_{\pm1.1}$ | $\mathbf{48.6}_{\pm2.3}$ | $\underline{49.5}_{\pm2.5}$ | $\mathbf{43.1}_{\pm2.1}$ | $\mathbf{45.0}_{\pm3.0}$ | **48.1** |
| *Five-Shot Classification* | | | | | | |
| CLIP (ViT-L/14) | $58.3_{\pm1.7}$ | $57.8_{\pm2.6}$ | $66.1_{\pm2.4}$ | $43.9_{\pm2.9}$ | $43.5_{\pm1.6}$ | 53.9 |
| SigLIP | $64.3_{\pm2.2}$ | $60.0_{\pm2.7}$ | $\underline{71.6}_{\pm1.4}$ | $46.2_{\pm1.8}$ | $42.6_{\pm1.7}$ | 56.9 |
| Supervised-IN21K | $51.3_{\pm2.3}$ | $48.5_{\pm2.6}$ | $60.5_{\pm3.0}$ | $39.3_{\pm3.4}$ | $41.9_{\pm1.6}$ | 48.3 |
| DINOv3 | $\underline{66.3}_{\pm3.3}$ | $\underline{63.2}_{\pm3.1}$ | $\mathbf{74.8}_{\pm1.5}$ | $\mathbf{59.9}_{\pm3.7}$ | $\underline{53.8}_{\pm2.4}$ | 63.6 |
| BioTrove-CLIP | $47.6_{\pm3.0}$ | $46.7_{\pm2.1}$ | $62.2_{\pm1.1}$ | $41.1_{\pm1.6}$ | $41.8_{\pm2.4}$ | 47.9 |
| BIOCLIP | $62.5_{\pm1.7}$ | $57.0_{\pm3.2}$ | $63.2_{\pm3.8}$ | $44.6_{\pm1.7}$ | $44.0_{\pm0.5}$ | 54.3 |
| BIOCLIP 2 | $\mathbf{73.4}_{\pm2.9}$ | $\mathbf{73.4}_{\pm0.9}$ | $70.7_{\pm1.6}$ | $\underline{59.4}_{\pm2.8}$ | $\mathbf{61.8}_{\pm1.3}$ | **67.7** |

## H  Biological Visual Evaluation Details

Instead of training specialist models, we treat the evaluated models as frozen visual embedding extractors. Standard machine learning algorithms are applied on top of the acquired visual embeddings to predict the corresponding labels. Such a design is adopted to evaluate the quality of the embeddings while avoiding the influence of complicated optimization loops. In the following, we introduce the details of the adopted benchmarks and the evaluation algorithms. All the experiments are conducted with 1 NVIDIA A100 GPU, and the running time for each task is within 30 minutes.

**FishNet.** FishNet focuses on recognizing, locating, and predicting species and their functional traits [16]. Specifically, 94,532 images are collected with annotations of habitat, ecological role, and nutritional value. In this work, we mainly focus on the prediction of habitats and ecological roles, involving 9 groups of binary labels (*e.g.*, whether the fish can live in freshwater). Following the practice in the original paper, we train a two-layer linear classifier with binary cross-entropy loss to predict the 9 labels. We count a correct prediction only if all the 9 labels are predicted correctly for the sample. The original train-test split is adopted, where 75,631 images are used in training, and the remaining 18,901 images are used for testing. This task evaluates whether the embedding distribution of different species is aligned with their ecological relationships.

**NeWT.** NeWT comprises 164 binary classification tasks in the natural world [18]. The tasks include appearance, gestalt, context, counting, and behavior concepts. In each task, 50-100 images are assigned per class per train-test split. After extracting visual embeddings, we apply a support vector classifier for each of the tasks. The average accuracy is reported across all the evaluated tasks.

**AwA2.** AwA2 consists of 37,322 images of 50 animal classes, with annotations of 85 numeric attribute values for each class [17]. The dataset can be used for testing the capability of attribute-based classification and zero-shot transfer learning of trait prediction. In this work, we mainly focus on the transfer learning scheme. The training set of 45 classes is used to train an attribute classifier, and the remaining 5 unseen classes are used for testing. Similar to FishNet, we incorporate a linear classifier on top of the extracted features to predict the binary labels for the 85 attributes. The average F1 score over all the attributes is reported for this benchmark.

**Herb. 19.** Herbarium 19 is a task for discovering new species [19]. Specifically, given the images of known and unknown species, the model is required to predict the labels for both of them. As there are no fixed labels for unknown species, the task is implemented with a form of semi-supervised clustering [85]. Given the total number of species, a semi-supervised K-means algorithm is conducted on top of the extracted embeddings to cluster images. Clustering accuracy is calculated for the predictions following the original practice [85].

**PlantDoc.** PlantDoc is a dataset targeting the incorporation of computer vision for scalable and early plant disease detection [20]. 2,598 images of 13 plant species and up to 17 classes of diseases are collected in uncontrolled natural settings. We evaluate the model on PlantDoc in a few-shot learning style. One image per class is randomly selected from the original training split as the support set, and SimpleShot [86] is employed to predict the class labels for the test set. Accuracy over all the testing samples is reported as the performance.

**Life stage-Diff/Align.** We use the data of Age tasks from NeWT [18] to construct the Life stage-Diff/Align benchmark, where images are labeled with juvenile and adult classes. Specifically, the differentiation tasks aim to separate the binary appearance variations within each species. Conversely, in the alignment task, we train a species classifier with juvenile images while testing it using adult images. It requires the embeddings of two variations within one species to be closer than the embeddings of different species. For both of the tasks, we incorporate a support vector classifier to predict the corresponding labels. Ideally, after extensive contrastive training, the embeddings of different variation classes are expected to collapse to the species prototype. However, we demonstrate that the intra-species variations are well preserved in the embedding space of BIOCLIP 2.

**Sex-Diff/Align.** NABirds consists of 48,000 images from 400 bird species [61]. Sex and life stage labels are provided for those species with large appearance variations. We manually examine the images and select 81 species with male-female differences to construct the Sex-Diff/Align benchmark, where 13,624 images are used in total. For the differentiation task, we filter out 20 male images and 20 female images per species as the test set. Among the remaining 10,384 images, we filter out at most 20 male and 20 female images for training. For the alignment setting, we use the images of female birds to train a species classifier and use the male images for testing. Similar to the life stage benchmark, we use a support vector classifier for both of the sub-tasks.

# I  Data Processing Details

In this section, we provide more details on the data curation process.

## I.1  Taxonomic Standardization

When GNVerifier returns a result from a query, our taxonomic alignment package combines it with the input taxonomic hierarchy, along with the query parameter, to form a resolution attempt. The query response, based on the most specific taxonomic term available in the input data, along with the remaining input hierarchy (entry's resolution attempt), is then matched against pre-defined profiles. The algorithm uses these three components to fit against the series of profiles to determine whether a confident resolution is found or if an alternative query strategy is needed (such as using a different query term or data source), iterating until a match is made or alternative approaches are exhausted.

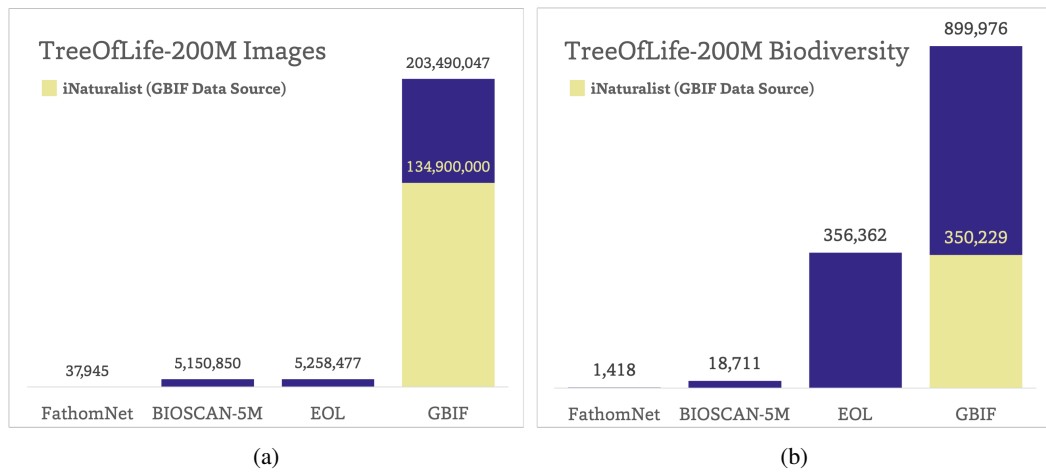

(a)                                    (b)

Figure 11: The number of images (a) and unique 7-rank taxa (b) by core data provider in TREEOFLIFE-200M. iNaturalist [42] is the largest GBIF data source (by image count) and the data provider used by BioTrove [14], so it is included here for reference.

Table 8: Top taxa contributors in GBIF [38]. Here, "Distinct Taxa" is used to describe 7-rank taxonomic labels only provided to GBIF by that publisher, while "Total Taxa" is the total number of unique taxa labels within that publisher. Observe that, following iNaturalist [42], the next three most diverse sources are museums, which, together, account for more distinct taxa than iNaturalist alone, despite having significantly fewer images overall.

| Publisher (GBIF Data Source) | Distinct Taxa | Total Taxa | Images | % GBIF Taxa |
|---|---|---|---|---|
| iNaturalist.org | 81, 671 | 350, 229 | 134, 877, 019 | 9.07% |
| Natural History Museum | 38, 535 | 148, 637 | 3, 603, 256 | 4.28% |
| Museum national d'Histoire nat. | 37, 201 | 258, 766 | 6, 185, 477 | 4.13% |
| Naturalis Biodiversity Center | 25, 421 | 244, 697 | 5, 282, 428 | 2.82% |

Table 9: The distribution of unique taxa and the covered images at different taxonomic ranks in TREEOFLIFE-200M. 92.26% images in TREEOFLIFE-200M have taxonomic labels specific to the species level.

| Rank | Total Images | Unique Taxa | Percent of Dataset |
|---|---|---|---|
| Species | 197,382,628 | 867,455 | 92.26% |
| Genus | 206,160,396 | 135,380 | 96.36% |
| Family | 207,489,189 | 13,790 | 96.99% |
| Order | 210,063,485 | 1,683 | 98.19% |
| Class | 211,236,966 | 382 | 98.74% |
| Phylum | 212,416,362 | 127 | 99.29% |
| Kingdom | 213,932,022 | 11 | 100.00% |

If all attempts ultimately fail to match a profile, the original input data is used for the entry's label in the final output. After hierarchical resolution, common names are annotated. For each entry, the common name corresponding to its most specific resolved taxonomic term is selected, when available, using the preferred English vernacular names from the GBIF Backbone Taxonomy [87]. In the output, the proportion of records changed for each data source is EOL: 98.7%, FathomNet: 16.0%, BIOSCAN: 11.8%, and GBIF: 0.3%. The notably low modification rate for GBIF reflects our taxonomic alignment package's preference for the GBIF Backbone Taxonomy as its primary reference. After standardization, we summarize the image number and taxa number from each source in Figure 11. We also report the distinct taxa and image number provided by top publishers from GBIF in Table 8. We provide the source code for this part in TaxonoPy.

As claimed in §3.3, not all taxa are specific to the species level. We summarize the detailed taxa distribution in Table 9. Although some ambiguous labels were mapped up to higher taxonomic ranks, our overall dataset still predominantly comprises images confidently labeled at the species level. Specifically, out of the total 214M images, approximately 92.26% retain species-level labels.

## I.2   Image-quality Screens

At download, all images were checked to be sufficiently large (224 pixels or more on the shortest side), and resized, where needed, so that the largest side does not exceed 720 pixels. The code, support set class embeddings, and further details of the image-quality screens and the subsequent duplicate control are provided in our dataset repository, `TreeOfLife-toolbox`.

### I.2.1   Museum Specimen Processing

Museums often consider multiple specimens of the same species to be connected instead of differentiating them (as they are often collected from the same or similar location). Thus, these occurrences often include images of their metadata, duplicates of the same or different specimens under one occurrence ID, or less informative images. Some common complications to consider when working with museum specimen images that influenced our processing are that

1. Plant and fungi specimens may be stored in envelopes or folders. These are often photographed and digitized under the same occurrence ID as the image of the specimen itself, thus creating extraneous images we do not want to include in training. They also sometimes pair this with living specimen images (which would be worth keeping as well). See Figure 12, which demonstrates variety within a single occurrence for fungi specimens. Plant specimens have a secondary confounding factor in that they are often pressed for preservation with their metadata on the page (see Figure 13).
2. Fish, worms, and similar will be stored in jars. A single jar is considered an occurrence, so only one specimen may be photographed—perhaps at multiple angles—the jar may be photographed, multiple specimens photographed, etc. In any of these cases, the images will all be labeled with the same metadata that does not include this context.
3. For animal specimens, both of the above cases may occur: there may be simply a close-up of the tag (similar issue to envelopes with plants and fungi). There may also be multiple specimens photographed within the same occurrence (as noted with 2 about fish and worms). Museums often consider multiple specimens of the same species to be connected instead of differentiating them. This is two-fold, in that they are generally, collected from the same or similar location at or about the same time, and a specimen is kept as a representative of its species, so there is not a clear need for distinguishing between them. We also see multiple views of the same specimen within an occurrence. Examples in Figure 14.

In order to appropriately separate these images, we treated them as museums do, specifically by first dividing them into 11 collection areas (Fungi, Insect, Invertebrate Zoology, Microbiology, Plant, Uncategorized, and five classes of Vertebrate Zoology: Amphibians, Birds, Fishes, Mammals, Others) inspired by the Smithsonian Institution's categorical subdivisions for their biological museum collections. From here, we further divided each category based on its image type (e.g., fossil or preserved specimen, as specified in GBIF metadata).

For each museum specimen category, we manually curated a small "support set" of representative specimen and non-specimen images. We embedded these examples using CLIP (ViT-L/14@336px) [7]; we chose not to use BIOCLIP since museum specimen labels were not filtered from its training data (e.g., EOL contains many museum specimen images). To capture intra-class diversity, we ran K-Means on each support set and retained the resulting cluster centers. During classification, we L2-normalized both the input image's embedding and each center, then assigned the image to the nearest center in Euclidean space. This processing was applied to all museum specimen images identified within GBIF. The support set embeddings are included in `TreeOfLife-toolbox`.

### I.2.2   Camera Trap Images

Some occurrences include large-volume camera-trap sequences, with up to 10,000 images. These occurrences have a single taxonomic label applied across all images, though there are different taxa (*e.g.*, the first few frames may have a duck, while later images have a swan, another a goose, but the label for all is a duck). To reduce the risk of introducing such noise while still capturing relevant biodiversity, we filter the dataset to include only occurrences with 15 images or fewer. We then use MegaDetector [47, 48] to filter "empty" frames. In creating a dataset to train a foundation model on the entire tree of life, it is prudent to avoid introducing too many plant images labeled as animals.

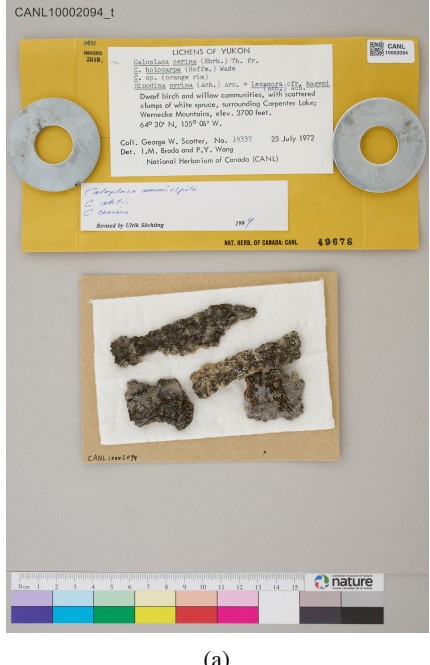
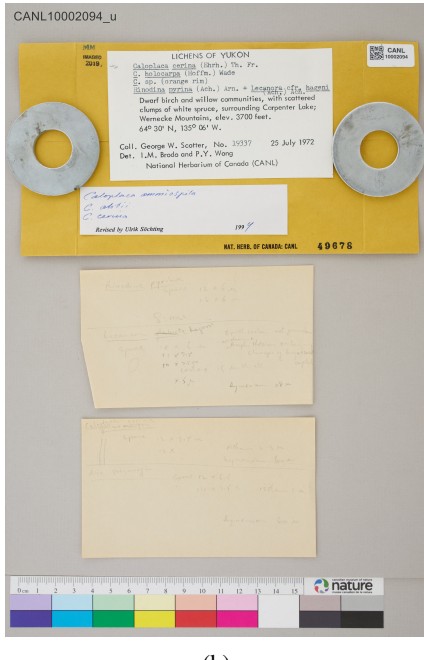

(a)            (b)

Figure 12: These images all belong to GBIF Occurrence 2301948912 [88], and thus share metadata. (a) Contains the specimen (*Caloplaca cerina*), (b) contains the file metadata from the specimen folder; only (a) should be retained. Collected in Canada by ©Canadian Museum of Nature (licensed under CC BY-NC 4.0).

### I.2.3 Citizen Science Images

Similarly, citizen science occurrences may contain many images assigned a single label. For each occurrence, we thus embedded the images using BIOCLIP [13], standardized them, calculated the pair-wise cosine similarity, and used the mean to indicate the occurrence's "distinctness". In these instances, we identified three broad categories into which to further subdivide them for reduction:

1. Mixed occurrences: multiple species all labeled as just one of them (low similarity). These were treated as noise and discarded.
2. Multiple image occurrences of the same species: observations suggest these are creature close-ups and environment images under a single occurrence. They may also be larger groups with close-ups. For these, we randomly sample down to 5 images.
3. Camera trap images: likely images collected in backyards (high similarity). Occasionally, people upload images from camera traps to citizen science platforms like iNaturualist (also ex: MammalWeb, a citizen science platform for camera trap images). These are processed under camera trap protocols described above.

The final step in our quality control processing pipeline was to filter out identifiable images of people from the training data. This was done by running MTCNN [49] on all images downloaded from GBIF and EOL.

### I.3 Duplicate and Leakage Control

GBIF and EOL are large biodiversity data aggregators, sourcing images that are also used in biological benchmarks such as iNat21 [36] and NeWT [18]. Due to metadata provenance complications, the images in these test sets cannot easily be matched to their sources or copies downloaded from other sources. To prevent the introduction of test data sourced from iNaturalist or "other sources online", we perform two content deduplication steps. The first is to run MD5 hashes on all downloaded images. During our download, these hash sums are used to distinguish images that have already been downloaded. We record the hash of both the original image and our resized image. These were used to ensure that Rare Species [59] images (sourced from EOL) were not included in the training data.

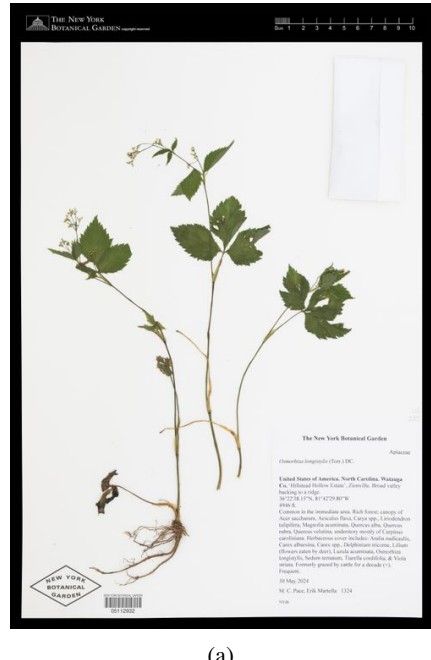
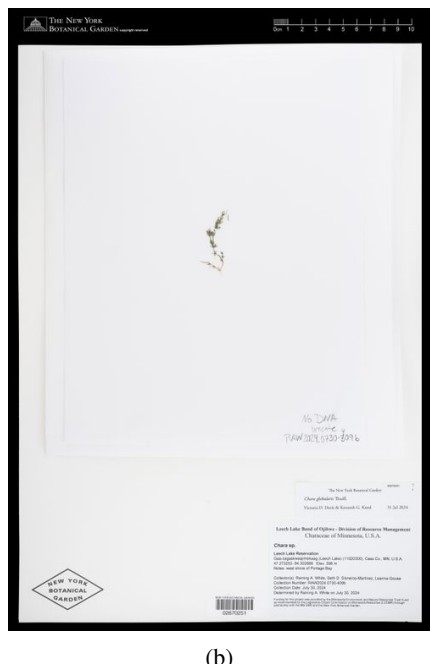

| (a) | (b) |
|:---:|:---:|

Figure 13: Both images contain specimens included in the dataset, but their dominance in-frame is quite different. The pressed flower sheets are the same size, but (a) *Osmorhiza longistylis* is dominant in the frame, while (b) *Chara globularis* covers less pixel area than the text. These images belong to GBIF Occurrence 5132787320 and GBIF Occurrence 5135831095 [88], respectively, and were both collected in the United States of America by The New York Botanical Garden (licensed under CC BY 4.0).

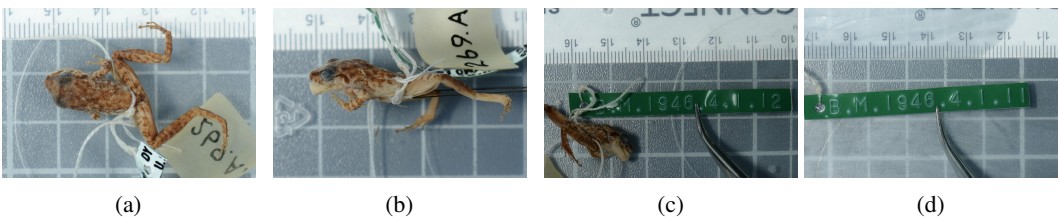

|  (a)  |  (b)  |  (c)  |  (d)  |
|:-----:|:-----:|:-----:|:-----:|

Figure 14: These images all belong to GBIF Occurrence 1056329684 [89], and thus share metadata. (a) through (c) all contain the specimen (*Pristimantis zeuctotylus*), though (c) is primarily focused on its label or tag. Meanwhile, (d) is the label for another specimen of the same species from the collection; there are multiple views of this specimen as well. (c) provides an alternate view and can be retained, while (d) should be removed.

Traditional hash sums are highly sensitive to small changes–a one-pixel difference between two images will produce a different hash. Hence, we applied perceptual hashing [PDQ 50], with distance less than 10, to identify training images that may be in our desired test sets that could not otherwise be filtered out (*i.e.*, by MD5 hash sum or through metadata). Note that PDQ hash evaluation was only run on GBIF citizen science images and EOL images sourced from Flickr since they are not reliable for museum specimens.

**IUCN Red List coverage.** According to the most recent IUCN Red List assessment[52], TREEOFLIFE-200M contains images of 69.5% (55,512) of all IUCN-assessed species in categories characterized as rare species or data deficient. This coverage was determined by applying our taxonomic alignment package to the IUCN taxonomic data to enable direct comparison with our standardized dataset. TREEOFLIFE-200M demonstrates particularly strong representation of threatened species, with 77.1% coverage across threatened categories (36,370 species), including 79.7% of Vulnerable species (14,038), 79.4% of Endangered species (15,190), and 68.4% of Critically Endangered species (7,142). Coverage extends to 81.7% of Near Threatened species (8,073) and 82.7% of species classified as Extinct in the Wild (67). Data-deficient species have a lower representation at 48.4% (11,002), likely reflecting the challenges in imaging and identifying the

species in this group. Notably, these species are designated in this category because there is not sufficient information about them for IUCN to evaluate their status; only $8\%$ of the 2.14M described species have been evaluated [51]. Thus, including $48\%$ of these species in TREEOFLIFE-200M, along with the threatened species coverage, establishes the approach to integrating diverse data sources used in TREEOFLIFE-200M as a valuable resource for conservation research, providing visual representation for a substantial majority of species prioritized for global conservation action.

## J   Author Contribution Statement

**Jianyang Gu** led the research project, ran the experiments, conducted analyses, and wrote the major part of the manuscript. For the dataset processing, he built the face detection pipeline to remove images with recognizable/identifiable people. He also retrieved the common names from GBIF backbone taxonomy alignment to `TaxonoPy` output.

**Samuel Stevens** constructed the initial benchmark for non-species classification tasks and significantly contributed to the paper writing. In addition, he also developed the content-based de-duplication pipeline based on PDQ-hash to identify test images that were contained in training data.

**Elizabeth G Campolongo** analyzed, evaluated, planned, organized, and documented the dataset effort. She developed the data processing approach by categories and supervised the tool development to retrieve and organize the dataset onto the HPC file system and of dataset curation. She curated camera trap test sets for benchmarking and Darwin's Finches for embedding analysis. She also contributed significantly to the dataset part of the paper writing.

**Matthew J Thompson** planned, organized, and supervised the development of tools to retrieve and organize the dataset onto the HPC file system. He developed and executed the taxonomic standardization (`TaxonoPy`) and webdataset ML-ready format conversion. He also contributed to the paper writing of the dataset part.

**Net Zhang** played an essential role in cleaning the noisy data. He dealt with the museum specimen images, citizen science images, and data de-duplication. Besides, he also processed and analyzed the downloaded data, primarily focusing on GBIF.

**Jiaman Wu** provided detailed advice on designing the training experiments for BIOCLIP 2. She also contributed to the dataset processing by initiating specimen label filtering and informing the taxonomic standardization for hemihomonym cases.

**Andrei Kopanev** developed the `distributed-downloader` tool to retrieve and organize the original images in the HPC file system and converted the dataset into ML-ready format (webdataset).

**Zheda Mai** provided constructive advice on designing the experience replay of CLIP training data.

**Alexander E. White** provided expertise in consultation on the museum specimen image processing approach.

**James Balhoff** and **Wasila Dahdul** provided expertise in consultation on the algorithm for taxonomic hierarchy resolution relevant to `TaxonoPy` development.

**Daniel Rubenstein** and **Hilmar Lapp** were involved in regular meetings and gave valuable feedback on results and experiments.

**Tanya Berger-Wolf** and **Wei-Lun Chao** provided insightful discussions and directions throughout the development of the project. They offered constructive advice for both the training design and the model evaluation. They also provided detailed comments on the manuscript.

**Yu Su** is the senior lead that oversaw the project and was involved in every aspect. He conceived the idea of observing emergent properties from scaling hierarchical contrastive training. He gave insightful guidance in formalizing the research question and directions to verify the new capabilities of BIOCLIP 2. He provided critical feedback and helped shape the manuscript.

