# OpenReview forum: "BioCLIP 2: Emergent Properties from Scaling Hierarchical Contrastive Learning"
_NeurIPS.cc/2025/Conference — NeurIPS 2025 spotlight_

### Official Review · Reviewer_2dWB · 2025-06-12

**Clarity:** 3
**Significance:** 3
**Originality:** 2
**Rating:** 5
**Confidence:** 3

**Summary:**

This paper presents BioCLIP‑XL, a CLIP-based model that maps biological images and taxonomic labels into a shared embedding space. Trained with hierarchical contrastive learning on the new TreeOfLife‑200M dataset, it boosts zero-shot species classification accuracy by about 18% over the original BioCLIP. The authors also identify “emergent properties” in the embeddings: species naturally sort by ecological or functional traits, while subtle within‑species differences stay isolated in their own subspaces. Ablation tests and theoretical analysis confirm these effects and explain how they arise.

**Questions:**

1) Could you include visualizations of the sample distribution at each taxonomic rank to more clearly illustrate the dataset’s imbalance?

2) Given the data imbalance, might you consider reporting class‑averaged or balanced accuracy alongside top‑1 accuracy for a more comprehensive evaluation?

**Ethical Concerns:**

["NO or VERY MINOR ethics concerns only"]

**Final Justification:**

My questions are answered by the authors. I will keep my score of 5.

**Limitations:**

Yes.

**Paper Formatting Concerns:**

No obvious formatting issues were found.

**Quality:**

3

**Strengths And Weaknesses:**

Strengths:

1) The paper improves the hierarchical contrastive learning model based on BioCLIP, effectively improving the model's efficiency in utilizing hierarchical information.

2) The TreeOfLife-200M dataset is proposed, providing high-quality data resources for subsequent biological vision research
It is very valuable for in-depth analysis of emergent properties.

Weaknesses:

Aside from the dataset imbalance and incomplete theoretical analysis noted in the paper, another potential drawback is the simplicity of the text input: it only uses taxonomic labels (scientific or common names) and does not include ecological or appearance descriptions for each rank. Despite this limitation, due to data availability and missing fine‑grained descriptions, the ablation studies and downstream evaluations show that the model still generalizes well across tasks.

---

> ### Author Rebuttal · Authors · 2025-07-31
>
> Dear reviewer,
>
> We sincerely thank you for the detailed and constructive comments, and are glad that you acknowledge our effort in scaling data and empirical observations. Please find the response below:
>
> ***W1. Incorporation of ecological or appearance descriptions***
>
> Thank you for the insightful comment. We agree that the ecological or appearance descriptions will likely facilitate the model generalization, which have been investigated at a preliminary stage of BioCLIP-XL. However, it is extremely challenging to collect such fine-grained textual information at the scale of TreeOfLife-200M that covers 952K unique taxa. Many species lack Wikipedia pages, and the descriptions often exist only in specialized taxonomic literature or original species descriptions that are not readily accessible in a structured form. Even if the species-level descriptions are available, there are inconsistencies between overall descriptions and instance images (e.g., juvenile and adult images). We view integrating richer ecological and morphological descriptions as an important direction for future work, which we expect will further improve the model’s performance and utility.
>
> ***Q1. Sample distribution at each taxonomic rank***
>
> Thank you for the suggestion. Due to the rebuttal restriction regarding figures, we provide the image number distribution at different taxonomy levels in the following table. For each taxonomic rank, we report the mean, median, minimum, and maximum image number per taxon. The largest taxon at each rank is also attached following the maximum image number.
>
> | Rank | Mean images per taxon | Median images per taxon | Min / Max images per taxon |
> | --- | --- | --- | --- |
> | Species | 227 | 8 | 1 / 679,902 (Anas platyrhynchos) |
> | Genus | 1,522 | 21 | 1 / 1,333,687 (Carex) |
> | Family | 15,318 | 62 | 1 / 10,810,236 (Asteraceae) |
> | Order | 123,285 | 295 | 1 / 18,279,859 (Lepidoptera) |
> | Class | 549,904 | 221 | 1 / 76,466,776 (Magnoliopsida) |
> | Phylum | 1,672,569 | 236 | 1 / 99,472,339 (Tracheophyta) |
> | Kingdom | 19,448,365 | 184,809 | 70 / 102,591,559 (Animalia) |
>
> This imbalance reflects the natural prevalence and sampling biases typical in ecological data. We will include comprehensive visualizations of these distributions in the revised manuscript to better illustrate the dataset’s structure.
>
> ***Q2. Balanced accuracy***
>
> Thank you for the question. While the training data is imbalanced, most of our evaluation benchmarks feature balanced class distributions.
> * The Rare Species dataset has 400 classes, 395 of which have 30 unique images. The remaining 5 classes have 29, 29, 27, 27, and 21 images, respectively.
> * All the test sets from Meta-Album (Plankton, Insects, Insects 2, PlantNet, Fungi, PlantVillage, and Medicinal Leaf) have balanced classes, each with 40 images.
> * The Camera Trap datasets have a balanced class split. Each class in Desert Lion, ENA24, Island, Orinoquia, and OH Small Animals has 11, 56, 10, 12, 12 images, respectively.
>
> The only exception is NABirds, where the largest class has 120 images, while the smallest one has only 13 images. We thereby report the micro (overall) and macro (class-averaged) accuracy of NABirds below. As expected, macro accuracy is lower across all models due to the imbalance. BioCLIP-XL maintains a significant performance advantage over the other models on all the tasks. We will include the test set information and comprehensive evaluation in the revised manuscript.
>
> | Model/Accuracy | 0-shot | 1-shot | 5-shot |
> | --- | --- | --- | --- |
> | CLIP/Micro | 66.5 | 42.7$\pm$0.8 | 68.2$\pm$0.3 |
> | CLIP/Macro | 63.3 | 41.7$\pm$0.9 | 67.0$\pm$0.4 |
> | SigLIP/Micro | 61.7 | 39.9$\pm$0.9 | 64.2$\pm$0.3 |
> | SigLIP/Macro | 60.0 | 39.1$\pm$0.9 | 62.7$\pm$0.5 |
> | BioCLIP/Micro | 58.8 | 57.4$\pm$1.2 | 78.2$\pm$0.3 |
> | BioCLIP/Macro | 54.7 | 55.8$\pm$1.1 | 76.8$\pm$0.3 |
> | BioCLIP-XL/Micro | 74.9 | 82.4$\pm$1.1 | 92.4$\pm$0.2 |
> | BioCLIP-XL/Macro | 72.2 | 80.7$\pm$1.3 | 91.3$\pm$0.2 |

---

> > ### Comment · Reviewer_2dWB · 2025-08-04
> >
> > Thank you to the authors for the detailed and thoughtful rebuttal. My questions have been answered, and I will keep my original score unchanged.

---

> > > ### Author Response · Authors · 2025-08-05
> > >
> > > We sincerely thank the reviewer for the constructive comments. We will make sure to incorporate the discussion in the revised manuscript.

---

### Official Review · Reviewer_awBE · 2025-06-28

**Clarity:** 3
**Significance:** 2
**Originality:** 2
**Rating:** 4
**Confidence:** 4

**Summary:**

The paper applies large-scale contrastive vision-language models to classify a dataset comprising 214 million images of living organisms. The objective is species classification, along with additional biological visual tasks such as habitat classification and trait prediction.

**Questions:**

I have no further questions.

**Ethical Concerns:**

["NO or VERY MINOR ethics concerns only"]

**Final Justification:**

I thank the authors for their rebuttal and addressing my concerns.

**Limitations:**

Yes.

**Paper Formatting Concerns:**

To the best of my knowledge there are no major formatting issues in this paper.

**Quality:**

3

**Strengths And Weaknesses:**

**Strengths**
- The experiments leverage a large-scale dataset containing 214 million images, which is an impressive data scale that strengthens the validity of the reported results and supports robust model training.

- The study performs classification over 10 distinct tasks, demonstrating the flexibility and broad applicability of the proposed approach.

- The authors systematically investigate the impact of data scaling on model performance across these tasks, providing valuable insights into how increased data size translates into improvements in generalization and task transfer.

- The presentation of ideas and results is well-organized and clear, supported by thoughtfully designed figures and tables that effectively communicate key findings to the reader.

- The paper goes beyond empirical observations by introducing and proving a theorem that explains the ecological alignment across species while separating intra-species variation, offering a sound theoretical foundation for the proposed methods.

- The experimental evaluation includes thorough comparisons against several baseline models on multiple benchmark datasets, helping to validate the effectiveness and superiority of the proposed approach through comprehensive empirical evidence.

**Weaknesses**
- The classification tasks evaluated in the paper may be considered relatively trivial downstream tasks, as they primarily rely on morphological data (images) to distinguish between organisms and their associated feature attributes. This could limit the broader applicability of the conclusions to more complex or less morphologically distinct domains.

- In the BIOSCAN-5M project, which the authors themselves cite and leverage in their experiments, multimodal data were used in conjunction with a contrastive loss function to perform species classification. However, the current paper does not adequately compare its proposed approach to BIOSCAN-5M, nor does it sufficiently discuss how its architecture builds upon or improves over the BIOSCAN-5M framework.

- The primary novelty of the paper appears to be centered on the scale of the dataset used in experiments. While this scale is impressive, there is a risk that the contribution could be perceived as incremental if the architectural innovations or methodological advances are not clearly distinguished beyond simply scaling up data.

---

> ### Author Rebuttal · Authors · 2025-07-31
>
> Dear reviewer,
>
> We sincerely thank you for the detailed and constructive comments, and are glad that you acknowledge our effort in scaling data and empirical observations. Please find the response below:
>
> ***W1. Significance of morphological-based evaluation***
>
> Thank you for the thoughtful comment. While BioCLIP-XL is intentionally focused on morphological/visual signals, we do not view the morphology-based evaluation tasks as “trivial.” Fine-grained species classification is inherently challenging due to the subtle visual differences between closely related organisms. While some biodiversity research necessitates more detailed examinations, such as X-ray or genetic sequencing, BioCLIP-XL offers the opportunity to propose hypotheses and a more targeted direction based on the morphological information.
>
> Beyond classifying known species, we also explore the novel and more complex task of new species discovery. Using the TerraIncognita dataset [1] as an example, we evaluate the model to map the previously unseen species to the correct family based on 264K insect species names in the TreeOfLife-200M dataset as the vocabulary. BioCLIP‑XL substantially improves top‑1 family accuracy over strong vision‑language baselines:
>
> | Model | CLIP (ViT-L-14) | SigLIP (ViT-L-16) | BioCLIP-XL |
> | --- | --- | --- | --- |
> | Accuracy (%) | 30.0 | 47.5 | 97.0 |
>
> Although assigning the exact name ultimately requires expert review and genetic examination, BioCLIP-XL offers a convenient and accurate way for hierarchical placement. This capability helps narrow the search space, prioritize comparing candidates, and accelerate the discovery. We will revise the paper to include these new experimental results.
>
> [1] Chiranjeevi, Shivani, et al. "TerraIncognita: A Dynamic Benchmark for Species Discovery Using Frontier Models." arXiv preprint arXiv:2506.03182 (2025).
>
> ***W2. Comparison with the model in BIOSCAN-5M***
>
> Thanks for the question. We agree that BIOSCAN-5M is a valuable resource and a strong prior effort in multimodal species classification. Moreover, BIOSCAN-5M and BioCLIP-XL represent two orthogonal directions to improve biological foundation models—BIOSCAN-5M introduces extra DNA information, while BioCLIP-XL studies the data scaling.
>
> We agree and believe that the extra DNA information is critical and helpful for accurate species classification. However, as we focus on comprehensive coverage across the entire tree of life, it is challenging to acquire the DNA information for the involved 952K different taxa and 214M images. Therefore, the employment of image and taxonomic labels is based on scalability and compatibility with large-scale data. Consequently, BioCLIP-XL can be broadly applied in a variety of settings where DNA information is not easily accessible, such as historical data, citizen science images, and field observations. We will include this discussion in the revised version.
>
> ***W3. Contribution of the paper***
>
> Thank you for your question. We humbly believe that “novelty” is not limited to creating new approaches or architectures. Novelty and contributions can come from demonstrating and discovering new insights. In our humble opinion, many recent breakthroughs in AI and machine learning stem from scaling up the data and learning how to train from it. With that being said, scaling up these is by no means trivial.
>
> We also want to emphasize that TreeOfLife-200M is not a trivial aggregation of multiple data providers. As illustrated in Section 3.2 and supplementary material Section C, we design dedicated pipelines to unify taxonomic labels, filter out defective images, and remove duplicates. These efforts represent technical innovations in dataset construction and are essential to achieve a dataset of this size and the training of BioCLIP-XL.
>
> With the scaled dataset, BioCLIP-XL achieves significant performance improvement in distinguishing species. Furthermore, this work demonstrates that BioCLIP-XL leads to broader scientific advances. We show that BioCLIP-XL serves as a general-purpose biological foundation model, providing state-of-the-art performance across a range of visual tasks beyond species classification. These properties were formally analyzed in the paper and would not be observable at smaller scales. These analyses reveal that scaling and structured supervision lead to improved and interpretable representations, providing insights for future biodiversity research and broader scientific domains.

---

> > ### Comment · Reviewer_awBE · 2025-08-05
> >
> > I thank the authors for their rebuttal and their efforts to address my concerns.
> >
> > - I would like to clarify that I did **not** mean to imply that morphology-based evaluation tasks are “trivial.” On the contrary, I recognize that distinguishing between species based solely on images can be highly challenging. For this reason, I believe that incorporating additional data modalities (e.g., genetic data) could support more accurate species classification.
> >
> > - Furthermore, by referencing BIOSCAN-5M, I intended to suggest comparing the species classification results of the approaches proposed in BIOSCAN-5M with those in the current study to better evaluate the effectiveness of the proposed model.
> >
> > Overall, I would like to keep my initial score unchanged.

---

> > > ### Author Response · Authors · 2025-08-07
> > >
> > > Thank you very much for your follow-up.
> > >
> > > ***Q1. Incorporating additional data modalities***
> > >
> > > We fully agree that incorporating additional data modalities has the potential to further improve identification accuracy and scientific reliability. Our vision for BioCLIP-XL is to serve as a practical complement to genetic methods in research-grade species recognition. At the same time, BioCLIP-XL can also be useful in scenarios where it is unnecessary, unaffordable, or infeasible to use DNA-based methods, such as citizen science images, smart camera traps, autonomous drone navigation, etc. We will incorporate this discussion in the revised manuscript.
> > >
> > > ***Q2. Species classification comparison with BIOSCAN-5M***
> > >
> > > Thank you for your suggestion. In the original BIOSCAN-5M paper, Table 4 reports a 47.0% accuracy for DNA-based sequence models on the unseen test set. However, they didn't include image-based classification results. Therefore, we utilized the embeddings from their GitHub repository to calculate similarities between image/DNA embeddings and text embeddings. Below, we show the results obtained by the BIOSCAN model and BioCLIP-XL:
> > >
> > > | Model | BIOSCAN-5M (image) | BIOSCAN-5M (image + DNA) | BioCLIP-XL |
> > > | --- | --- | --- | --- |
> > > | Top-1 Accuracy | 35.2% | 41.6% | 58.5% |
> > >
> > > We note that the entire BIOSCAN-5M dataset (including validation and test sets) was used in BioCLIP-XL training, and the direct similarity calculation might not fully reflect the optimal performance of the BIOSCAN model. Therefore, the comparison should be viewed as approximate and primarily for reference.
> > >
> > > Thank you again for your constructive feedback and for helping us improve the clarity and rigor of our work.

---

### Official Review · Reviewer_A1KT · 2025-07-02

**Clarity:** 3
**Significance:** 3
**Originality:** 2
**Rating:** 4
**Confidence:** 4

**Summary:**

The authors introduce Bio-CLIP-XL, a powerful CLIP model trained on the mega-scale TREEOFLIFE-200M dataset, comprising 214 million image-text pairs of living organisms from sources including GBIF, BIOSCAN, and EOL. By achieving state-of-the-art performance on multiple benchmarks, this work represents a significant contribution to the machine learning community. The paper also provides insightful discussions on the model's training strategy to avoid neural collapse and its resulting emergent properties.

**Questions:**

1. This work represents a significant contribution to morphology-based species identification using deep learning. In modern biology, however, there is a strong trend towards using genomic sequencing (e.g., DNA barcoding) for this task, often seen as providing a more direct "ground truth". Could the authors elaborate on the specific scientific and practical advantages of their image-based approach? In what scenarios is a model like Bio-CLIP-XL superior to, or a necessary complement to, genomic methods for understanding biodiversity?

2. Could the authors clarify the composition of the 5 million images from the BIOSCAN dataset? The supplementary material suggests all 5M images were used, but the original BIOSCAN[1] dataset documentation indicates only ~473,000 images have expert-verified morphological labels, with others labeled via DNA barcoding.


   2.1. What was the labeling methodology for the ~4.5 million images that lacked expert morphological IDs?


   2.2. Could you also explain the reduction in class count from the 22,622 species in the source data to the 18,711 classes used in your final dataset?

3. Back to weakness 3. I am thinking a possibile case it could become the strength. What if, the text embeddings of different names are too similar in Bio-CLIP-XL may suggesting a taxnonmy merger? And this merge is happened in academia. If this is confirmed, the model can be a guide for taxnonmy revisement.

[1]. Gharaee, Z., Lowe, S. C., Gong, Z., Millan Arias, P., Pellegrino, N., Wang, A. T., ... & Chang, A. (2024). Bioscan-5m: A multimodal dataset for insect biodiversity. Advances in Neural Information Processing Systems, 37, 36285-36313.

**Ethical Concerns:**

["NO or VERY MINOR ethics concerns only"]

**Final Justification:**

I will keep my score of 4. Details in my reply.

**Limitations:**

See weakness.

**Quality:**

4

**Strengths And Weaknesses:**

Strengths
1. Scale and Diversity of the Training Data: The assembly of TREEOFLIFE-200M is a major achievement.

2. Robust Model Training and State-of-the-Art Performance: The paper demonstrates a robust and technically sophisticated training strategy, effectively leveraging large-scale computational resources. The resulting model, BIOCLIP-XL, sets a new state-of-the-art on a wide range of biological benchmarks, showcasing its capabilities as a powerful foundation model for the life sciences.

3. Avoided the Neural Collpsing often seen in constrast learning on fine grained data, and provided insightful discussion.

4. Insightful Theoretical Framework: The theoretical analysis, particularly Theorem 5.1, provides a compelling mathematical justification for the observed preservation of fine-grained features. It elegantly explains how the model learns to represent different factors of variation (species identity vs. intra-species traits) in orthogonal subspaces, providing a solid theoretical foundation for the empirical results.

Weakness
1. Questionable Foundation of Taxonomic Standardization: The decision to standardize all taxonomic labels against the GBIF Backbone Taxonomy as a single source of truth is a significant concern from a biological standpoint. While pragmatic for automation, GBIF is a secondary aggregator, not a primary taxonomic authority. Biological classifications are dynamic and constantly revised in peer-reviewed literature. By anchoring the entire dataset to a single, potentially lagging source, the model may have learned outdated species concepts, which could lead to critical errors in downstream applications (e.g., misidentifying recently split species with different conservation statuses).

2. Potential Mismatch with Core Scientific Needs for ecologists and conservationists: The model's exclusive focus on image identification ("what") overlooks the spatiotemporal metadata (location and date) that is often more critical for its target users. For ecologists and conservationists, core tasks like tracking species ranges, modeling habitats, or monitoring invasive species depend on knowing "where" and "when" an observation occurred. By discarding this essential context, the model's practical utility for a large portion of the biological community is significantly limited. The impact of the work on biology would be significantly increased if the dataset could provide high quality data on where and when the animals are spotted.

3. Fragility of the Text-Embedding-Based Prototypes: The theory relies on class prototypes being defined by text embeddings of taxonomic names. This creates a brittle system that is highly vulnerable to taxonomic revisions. For example, if a species is reclassified (a common occurrence), its text-based prototype could become instantly obsolete. A name change could cause a prototype to point to an entirely different concept, a species merger could cause two distinct prototypes to collapse into one, and a species split would leave the model with no prototype for the newly described entity. This tight coupling of the model's core structure to a non-permanent string of text could be a vulnerability.

---

> ### Author Rebuttal · Authors · 2025-07-31
>
> Dear reviewer,
>
> We sincerely thank you for the detailed and constructive comments, and are glad that you acknowledge the empirical and theoretical significance of this work. Please find the response below:
>
> ***W1. Taxonomic standardization solely based on GBIF backbone***
>
> Thank you for this insightful and important critique. We fully acknowledge that relying on the GBIF backbone taxonomy as a single taxonomic reference has potential limitations. As noted by the reviewer, biological classifications are inherently dynamic, and any taxonomy could be out of date with the ongoing scientific updates. Using GBIF provides a practical, scalable approach to unify taxonomic labels across vast datasets, accepting some degree of temporal lag as a trade-off for coverage and consistency.
>
> Built upon the unified taxonomy backbone, BioCLIP-XL illustrates capabilities beyond zero-shot classification. As illustrated in the reply to weakness 3, BioCLIP-XL learns generalizable visual features that can distinguish species and even intra-species variations without taxonomy labels. Moreover, as in the reply to question 3, BioCLIP-XL offers an opportunity to cross-validate the current taxonomic splits. Therefore, we humbly believe that the lagged taxonomy update won’t be a fatal limitation to BioCLIP-XL.
>
> ***W2. Lacking spatiotemporal metadata in the dataset***
>
> Thank you for the important observation. We agree that spatiotemporal metadata are critical for many ecological and conservation applications. However, incorporating such metadata at scale presents several challenges.
> * First, spatiotemporal data is often incomplete or unavailable for many images in large biodiversity datasets.
> * Second, for sensitive species (e.g., endangered or threatened), sharing precise location information raises ethical concerns regarding their protection.
> * Moreover, species ranges are changing. Relying strictly on expected location can limit practical utility in dynamic scenarios, such as poaching detection or invasive species detection.
>
> With that being said, lacking spatiotemporal information during training does not limit its use for application. BioCLIP-XL is designed as a biological foundation model that can be fine-tuned to suit specific needs, including spatiotemporal context. Additionally, people can acquire a potential species list given geolocation, which has been validated in the field study to facilitate more accurate species classification.
>
> ***W3. Dependency on text embedding of taxonomic names***
>
> Thank you for raising this important point. However, we would like to clarify that although BioCLIP-XL is trained with taxonomic labels, its learned visual-semantic space is not strictly bound to these specific text embeddings. Importantly, BioCLIP-XL demonstrates strong performance not only in open-set and zero-shot classification but also in few-shot settings. Given only one example image per class, BioCLIP-XL achieves 64.1% average accuracy across the 10 species classification benchmarks, outperforming its zero-shot setting (55.6%). This indicates that BioCLIP-XL does not solely rely on taxonomic labels to predict species. The visual embeddings can be used directly to construct new prototypes or support few-shot adaptation for newly described or reclassified species.
>
> Finally, we note that BioCLIP-XL’s visual representations enable applications beyond species classification. The illustrated visual biological tasks in the paper do not depend on taxonomic labels. Thus, while taxonomic changes are an inherent challenge in biological research, we do not see them as a fundamental limitation for BioCLIP-XL.
>
> ***Q1. The relationship between image-based and DNA-based methods***
>
> Thank you for this insightful question. We fully recognize that DNA barcoding is a more fundamental way to identify species. At the same time, image-based approaches such as BioCLIP-XL have distinct scientific and practical advantages that make them indispensable in many real-world scenarios.
> * **Scalability and data availability**.
> Genomic data collection is often limited by cost and accessibility. In contrast, image data can be collected at scale from diverse sources, including field cameras, citizen science platforms, and historical archives. For example, our dataset comprises images of 952K unique taxa—a scale that is currently unattainable for DNA-annotated samples, especially for wild populations and rare or elusive species.
> * **Accessibility and non-invasiveness**.
> Obtaining DNA samples requires specialized equipment, trained personnel, and often the physical collection or handling of organisms. However, this may not be feasible or ethical in many settings. In comparison, image-based methods allow rapid, non-invasive identification, making biodiversity monitoring more accessible to researchers and the general public.
> * **Complementary information to genomics**.
> Although DNA provides ground-truth identification, it does not capture all aspects relevant to biodiversity studies. For example, life stage, behavior, health, or phenotypic adaptations are often visually apparent but not easily inferred from genetic data alone. As demonstrated in the paper, BioCLIP-XL can distinguish between juvenile and adult individuals, complementing genetic approaches.
>
> ***Q2. Clarification of the incorporation of BIOSCAN-5M***
>
> Thank you for these questions. We clarify the following:
>
> *Q2.1: Labeling methodology for the images without expert morphological IDs*
>
> We used the labels as provided in BIOSCAN-5M. To ensure quality and consistency, we performed additional cleaning, including resolving residual ambiguous labels such as “Malaise####” by mapping them to higher taxonomic ranks where appropriate. This cleaning was done in close consultation with the BIOSCAN-5M dataset creators, who subsequently updated the dataset to improve label accuracy.
>
> *Q2.2: Explanation for class reduction*
>
> We are sorry that the discrepancy was not carefully explained in the manuscript. The 22,622 species in the BIOSCAN-5M paper refer to unique species-level entries. In our paper, we measure the number of unique 7-rank hierarchies instead of species. Additionally, we apply taxonomic standardization and alignment across all data providers, which resolves many irregular or placeholder labels. As a result, our resolved dataset includes 18,711 unique classes. We report in the supplementary material, line 64, that 11.8% of BIOSCAN-5M images were changed during the resolution.
>
> We will clarify these points in the revised manuscript.
>
> ***Q3. The potential of BioCLIP-XL as a guide for taxonomy revision ***
>
> It is an insightful comment. We believe that similarity between text embeddings of different species names in BioCLIP-XL could potentially indicate taxonomic mergers or revisions.
>
> For example, we observe cases where the text embedding of a mislabeled species (e.g., an insect labeled as a plant) clusters more closely with the correct group (other insects) than with its incorrect label (plants). While this is a different issue from taxonomic merger, it similarly shows how BioCLIP-XL’s embedding space reflects biological and semantic relationships that can guide taxonomy-related hypotheses.
>
> In all cases, we emphasize that these hypotheses should be validated with DNA barcoding and expert review. Nevertheless, BioCLIP-XL demonstrates the possibility to support taxonomic research by identifying potential revisions such as mergers or misclassifications.

---

> ### Comment · Reviewer_A1KT · 2025-08-02
> **Discrepancy between needs of biological researchers and research focous of machine learning researchers**
>
> Thank you for the detailed reply. I want to reiterate that I believe this is an amazing work from a machine learning perspective, particularly the analyses on preventing neural collapse, which is the primary reason for my positive evaluation.
>
> However, I must clarify that my core concerns remain unaddressed. I believe these concerns stem from a fundamental discrepancy between the research focus of the machine learning community and the practical needs of biological researchers. As I see it, the primary audience for a tool like BioCLIP-XL consists of biologists seeking a robust tool for taxonomy. To that end, I would like to restate my points, which I believe would significantly increase the paper's impact and adoption within the biological community.
>
> **1. The Need for a Fine-Tuning Tutorial**
>
> For many biologists, the primary concern is scientific accuracy at the species level. While achievements in zero-shot accuracy and generalizable features are impressive from a machine learning perspective, they are secondary to the need for taxonomically precise and verifiable identifications. We cannot use labels from sources like GBIF blindly, as taxonomic accuracy is paramount.
>
> The true value of BioCLIP-XL for our community may be as a powerful foundation model. A minimal, well-documented code tutorial demonstrating how to fine-tune the model with our own curated datasets (using updated nomenclature and verified images) would be invaluable. This would empower researchers to create downstream models that produce scientifically trustworthy results tailored to their specific needs.
>
> **2. The Scalability of Image-Based vs. DNA-Based Taxonomy for Research-Grade Identification**
>
> I would like to challenge the assertion of scalability, particularly when the goal is **research-grade** identification. The BIOSCAN dataset itself illustrates this point perfectly: it contains ~5 million DNA barcoded samples, but only ~460,000 with expert morphological labels required for scientific reporting. This highlights a counter-intuitive reality in biology: while images are easy to *acquire*, achieving a **research-grade** identification from them is *fundamentally difficult and slow*.
>
> A *research-grade* species identification from images is not a simple classification. It often requires multiple viewing angles, careful examination of minute morphological features against scientific literature, and ultimately, verification by a taxonomic expert. The true bottleneck is not data acquisition, but this rigorous expert annotation process.
>
> Conversely, while DNA can be more difficult to acquire, it provides a more direct and scalable path to a *ground-truth* taxonomic identification suitable for scientific studies. For large-scale research requiring high-confidence, species-level data, DNA barcoding is often the more reliable and scalable method for establishing taxonomy.
>
> **3. The Critical Importance of Spatiotemporal Metadata**
>
> I must respectfully push back on the decision to omit spatiotemporal data. The authors state that "species ranges are changing," and this is precisely the phenomenon that much of modern ecological and conservation research aims to analyze. Lacking this data severely constrains the utility of this work for biologists.
>
> Spatiotemporal data allows us to track the spread of invasive species, analyze distribution shifts in response to climate change, and identify areas in need of conservation attention. While we all agree on the need to protect location data for sensitive species, this can be managed through established methods like data aggregation or coordinate generalization, not just outright omission. Providing large-scale spatiotemporal data would dramatically increase interest from the biological and ecological research communities.
>
> **4. Clarification on Data Distribution Across Taxonomic Ranks**
>
> Your method of mapping ambiguous BIOSCAN labels (e.g., "Malaise####") to higher taxonomic ranks implies that the vast majority of the ~4.5 million images lack species-level labels. Furthermore, your cleaning process, such as revising *Orthocentrus Malaise5162* to *Orthocentrus spp.*, actively reduces the number of species-level samples. (Note: Revision of *Orthocentrus Malaise5162* to genus-only like *Orthocentrus spp.* is correct. However, revising it to *Orthocentrus Malaise* would be problematic, as the original label explicitly marks it as a distinct morphospecies.)
>
> For the sake of transparency and to allow researchers to understand the model's training basis, it is crucial to provide a table explicitly reporting the distribution of images across the taxonomic hierarchy (i.e., number of images with confident labels at the species, genus, family, order, etc., levels). This transparency is essential for evaluating the model's capabilities and limitations.
>
> I hope these points could help to bridge the gap between this powerful model and its intended user base.

---

> ### Author Response · Authors · 2025-08-05
> **Reply to the comments (2/2)**
>
> ***Q4. Clarification on Data Distribution Across Taxonomic Ranks***
>
> Thank you for highlighting this important point. We apologize that this information was not included in the original manuscript.
>
> We want to first clarify that for Orthocentrus Malaise5162 (and for similar cases of provisional labels), we omitted the species label and resolved higher ranks with the genus as Orthocentrus. It is true that the samples lack species labels. However, this sacrifice helps to make sure that an instance from BIOSCAN-5M wouldn’t be marked as unique from another instance of the same species from EOL or GBIF.
>
> Although some ambiguous labels were mapped up to higher taxonomic ranks, our overall dataset still predominantly comprises images confidently labeled at the species level. Specifically, out of the total 214M images, approximately 92.26% retain species-level labels. Below, we provide a detailed distribution table across all taxonomic ranks:
>
> | Rank | Total images | Unique taxa | Percent of dataset |
> | --- | --- | --- | --- |
> | Species | 197,382,628 | 867,455 | 92.26% |
> | Genus | 206,160,396 | 135,380 | 96.36% |
> | Family | 207,489,189 | 13,790 | 96.99% |
> | Order | 210,063,485 | 1,683 | 98.19% |
> | Class | 211,236,966 | 382 | 98.74% |
> | Phylum | 212,416,362 | 127 | 99.29% |
> | Kingdom | 213,932,022 | 11 | 100.00% |
>
> We will explicitly incorporate this comprehensive taxonomic breakdown into the revised manuscript, ensuring full transparency of our dataset composition.
>
> Again, we want to sincerely thank the reviewer for raising these important questions. We believe that these questions are critical for further improving the paper quality and making BioCLIP-XL a practical tool. We hope that the reply addresses your concerns. Please kindly let us know if you have any further questions.

---

### Decision · Program_Chairs · 2025-09-17

**Decision:**

Accept (spotlight)

**Comment:**

This paper presents a new large scale biological organism image dataset (TreeOfLife-200M dataset) and a model (BioCLIP-XL) trained on these biological data with contrastive learning. The paper also presents several analysis and experiments on the learned embedding space of BioCLIP-XL.

All the reviewers are positive about the paper, valuing the new large scale dataset (with more than 2M images), the state-of-the-art performance of the BioCLIP-XL model on different biological tasks, and the theoretical insights discussed in the paper. The Author-Reviewer discussion was active, and the authors provided satisfactory answers to the questions of the reviewers, making the reviewers to keep their original positive score. Regarding weaknesses, there was some discussion on the discrepancy between the needs of biological researchers and the research focus of machine learning researchers. We recommend the authors to add some discussion regarding this aspect in the final version of the paper, to encourage future research that better aligns with the needs of biological researcher.

I recommend this work for spotlight due to its potential impact in a relatively underexplored area. The work provides a large scale curated dataset, a trained embedding space, and extensive analyses and experiments on the resulting embedding space, providing valuable insights and tools for the research community. Assuming that the authors will release the dataset and the trained model, we believe these resources will open up new directions for future research in biological vision.